# Modelling braided river morphodynamics using a particle travel length framework

Alan Kasprak[1*], James Brasington[2], Konrad Hafen[1,3], Richard D. Williams[4], and Joseph M. Wheaton[1]

[1]Department of Watershed Sciences, Utah State University, Logan, Utah, 84322, USA
[2]Te Waiora, Institute for Freshwater Management, University of Waikato, Hamilton 3240, New Zealand
[3]Water Resources Program, University of Idaho, Moscow, Idaho, 83844, USA
[4]School of Geographical and Earth Sciences, University of Glasgow, Glasgow, G12 8QQ, UK

*Now at US Geological Survey, Southwest Biological Science Center, Grand Canyon Monitoring and Research Center, Flagstaff, Arizona, 86001, USA

Correspondence to: Alan Kasprak (akasprak@usgs.gov)

**Abstract.** Numerical models that predict channel evolution are an essential tool for investigating processes that

occur over timescales which render field observation intractable. The current generation of morphodynamic models, however, either oversimplify the relevant physical processes, or in the case of more physically-complete CFD based codes, have computational overheads that severely restrict the space-time scope of their application. Here we present a new, open-source, hybrid approach that seeks to reconcile these modelling philosophies. This framework combines steady-state, two-dimensional CFD hydraulics with a rule-based sediment transport algorithm

to predict particle mobility and transport paths which are used to route sediment and evolve the bed topography. Data from two contrasting natural braided rivers (Rees, New Zealand and Feshie, United Kingdom) were used for model verification, incorporating reach-scale quantitative morphological change budgets and volumetric assessment of different braiding mechanisms. The model was able to simulate eight of ten empirically observed braiding mechanisms from the parameterized bed erosion, transport, and deposition. Representation of bank

erosion and bar edge trimming necessitated the inclusion of a lateral channel migration algorithm. Comparisons between simulations based on steady effective discharge versus event hydrographs discretized into a series of model runs were found to only marginally increase the predicted volumetric change, with greater deposition offsetting erosion. A decadal-scale simulation indicates that accurate prediction of event-scale scour depth and subsequent deposition present a methodological challenge because the predicted pattern of deposition may never 'catch up' to

erosion if a simple path-length distribution is employed, thus resulting in channel over-scouring. It may thus be necessary to augment path length distributions to preferentially deposit material in certain geomorphic units. We

anticipate that the model presented here will be used as a modular framework to explore the effect of different process representations, and as a learning tool designed to reveal the relative importance of geomorphic transport processes in rivers at multiple timescales.

## 1 Introduction

The dearth of morphodynamic models that resolve the bar-scale morphology of braided, gravel-bed rivers remains a first-order weakness in fluvial geomorphology. Indeed, since prediction is generally perceived to be the pinnacle of scientific enquiry, the limitations associated with existing modelling frameworks that can be used to inform management and natural hazard assessment of braided rivers restrict the contribution that our science can make to addressing societal needs (Wilcock and Iverson, 2003). Focusing model development on braided rivers is paramount since, in principle, if a numerical framework can predict the evolution of multiple channels and bars during high-flow events in these multi-thread rivers then the same framework should be transferable to single-thread rivers. Major progress has been made in applying contemporary measurement technologies to quantify the morphodynamics of braided rivers at the timescale of individual to sequences of high-flow events ($10^{-2}$-$10^0$ years; Lane et al., 2003; Wheaton et al., 2013; Williams et al., 2011; Lallias-Tacon et al., 2014; Williams et al., 2015). Prediction is, however, needed at the mesoscale (broadly $10^0$-$10^2$ km and $10^1$-$10^3$ years; Brasington and Richards, 2007), which exceeds the scale of feasible field-based monitoring campaigns. There have been considerable advances in modelling realistic bar-scale morphology of fine-grained braided rivers at the mesoscale (Nicholas, 2013a; Schuurman et al., 2013; Schuurman et al., 2015) but models that resolve bar-scale morphology of gravel-bed rivers at timescales greater than single high-flow events (Williams et al., 2016b) remains challenging. Furthermore, while 'physics-based' (Nicholas, 2013b) approaches to simulation have been successful for fine-grained rivers, such approaches have high computational overheads necessitating the use of high performance computing resources and restricting their use for exploratory applications (e.g. Nicholas et al., 2013a). The additional computational demand of simulating sediment mixtures, combined with the extant problems of spatial parameterization of sediment character, limit the scope of comparable 'physics-based' approaches to gravel-bed river simulations. There is therefore a need to develop an alternative, computationally efficient morphological modelling framework, capable of reproducing realistic bar-scale morphodynamics of gravel-bed braided rivers over geomorphologically meaningful timescales ($10^1$-$10^3$ years).

A variety of spatially-distributed morphodynamic modelling frameworks have been developed to address this problem and are reviewed in depth by Williams et al. (2016a). Current approaches have been broadly assigned

into one of two philosophical categories. The first involves simplifying and abstracting physical processes using a set of rules or simplified algorithms, giving rise to so-called 'reduced complexity' models, often in the form of cellular automata models (RC/CA; Murray and Paola, 1994; Coulthard et al., 2002; Thomas and Nicholas, 2002). These rule-based models offer high computational efficiency, allowing calculations over wide spatial and long

temporal scales (e.g., Nicholas and Quine, 2007; Thomas et al., 2007; Ziliani et al., 2013), but often at the expense of morphological fidelity to any particular system, instead producing self-organization and generalized behaviours of a given channel type (e.g., Murray and Paola, 1994; Murray, 2003). RC/CA models are particularly well suited for investigating the generalized morphodynamic response of channels under shifting boundary conditions (Thomas et al., 2007). The alternative second subset of models are widely referred to as 'physics-based', and are

driven by computational fluid dynamics schemes (CFD; Bates et al., 2005) typically involving two-dimensional approximations of solutions to the Navier-Stokes equations. Bed shear stress is used as a measure of the friction force imposed by flow to scale bedload transport. The direction and distance of bedload transport is calculated based upon the flow field and parameterizations are employed to represent gravity driven sediment transport, particle settling and remobilization effects. The Exner equation (Paola and Voller, 2005) is then used to calculate

bed elevation change from local sediment flux divergence. This 'physics-based' approach to calculating morphodynamic evolution comes at the cost of significantly increased computational overheads, particularly for graded sediment. Nonetheless, physics-based models have been widely applied to simulate fluvial systems (Mosselman et al., 2000; Rinaldi et al., 2008; Kleinhans, 2010) and event-based, graded sediment simulations reproducing reach-scale morphodynamics of natural rivers are beginning to emerge (Williams et al., 2016b).

Moreover, such models have been used to shed light on how the morphodynamics of braided rivers are influenced by sediment heterogeneity (Sun et al., 2015; Singh et al., 2017) and vegetation (Crosato and Saleh, 2011; Li and Millar, 2011; Nicholas et al., 2013a; Iwasaki et al., 2017). Despite this progress, mesoscale physics-based simulations require considerable computational resources and can diverge from predicting feasible morphology (Schuurman et al., 2015). Beyond this classic dichotomy, a third approach to morphodynamic simulation draws

on particle-based methods from granular physics (Frey and Church, 2012). This approach is however, computationally intensive and the absence of appropriate upscaling methods limits its application to patch-scale investigations of sediment entrainment, transport and deposition (e.g., Escauriaza and Sotiropoulos, 2011; Nabi et al., 2013).

The lack of morphodynamic modelling approaches that are optimized effectively for use at the mesoscale presents the opportunity for an alternative framework. An attractive approach is to couple flow routing predictions based

upon the well-understood and robust shallow water wave equations (Nicholas et al., 2012; Williams et al., 2013) with a simplified, empirically-derived rule set for sediment transport, which in turn is used to update the bed topography. This schema has the advantage of decoupling the fastest components of the system (the hydrodynamics) from the slowest (evolution of the bed). Furthermore, an approach to sediment transport modelling based on particle path lengths, the characteristic distance travelled by sediment particles during a flood (Davy and Lague, 2009; Furbish et al., 2016), has potential to simplify simulation of bedload transport representing the fundamental process that leads to bar formation and morphodynamics. Recent work indicates that morphology, bed structure, and texture play a key role in particle dispersion (Hassan and Bradley, 2017). Particle path length distributions have been found to take several forms in gravel-bed braided rivers, including exponential decay, or heavy-tailed distributions, marked by a large number of particles mobilized short distances downstream, resulting from floods that do not generate sufficient shear stress for transport across the braidplain width (Pyrce and Ashmore, 2003b). During floods which are competent across large areas of the braidplain, typical path length distributions exhibit peaks which correspond to the location of likely depositional sites downstream. Kasprak et al. (2015) and Pyrce and Ashmore (2003a,b) both noted that these depositional sites were most frequently the location of bar heads (e.g. flow diffluences) and as such, particle path length distributions could be readily constructed using morphometric indices that reflect the characteristic confluence-diffluence spacing.

This paper presents a new, hybrid morphodynamic modelling approach that employs two-dimensional CFD hydraulics resolved at the event scale and a rule-set for sediment transport that leverages hydraulic predictions to determine particle path lengths along which to erode and route sediment and subsequently evolve topography. Previous research on particle path length distributions has largely been conducted in gravel-bed braided rivers, and as such the model described here, termed MoRPHED (Model of Riverine Physical Habitat and Ecogeomorphic Dynamics), has been developed for such environments. Characteristic particle travel distances have, however, also been documented for single-thread channels (Pyrce and Ashmore, 2003b), so it may therefore be possible to apply the underlying theory to a wide variety of channel forms. That said, the approach is only appropriate where the extremes of the path length distribution are less than the length of the system to be modelled (i.e. for gravel- rather than sand-bed rivers).

Combining both dynamical (force-based hydraulics) and kinematic (motion-based prediction of particle transport) frameworks provides an effective compromise that incorporates the necessary physics to simulate the key driving forces and vectors of motion, but simultaneously offers a reduced complexity structure suitable for wide-area, long-

term morphodynamic modelling. This paper explores the degree to which MoRPHED (a) can capture the emergent properties of natural fluvial environments, (b) can maintain, but not necessarily form or produce, braided topography, and (c) exhibits sensitivity to contrasting process representations.

For transparency and ease of future development, the model presented here has been packaged into an open source code published at https://github.com/morphed/MoRPHED and a user interface (https://github.com/morphed/MoRPHED-Viewer). The novel contributions set out in this paper are: (i) a new, numerical morphodynamic modelling framework that combines a dynamics-based, CFD approach to predicting flow routing and a kinematics-based, particle travel length rule set for sediment transport and morphological
change; (ii) testing the modelling framework using data from two contrasting natural rivers, with sensitivity analyses to assess hydrograph discretization strategies and path length statistical distributions; and (iii) multi-scalar model verification using a plurality of rigorous validation approaches including reach-scale quantitative morphological change budgets and the pattern of contribution of different braiding mechanisms.

**2 The Model**

As with many previously-developed morphodynamic models, the model used here (MoRPHED v. 1.1) simulates hydraulics and uses these calculations to predict bedload transport and morphological change. This section details the methods used in in each of these components, along with ancillary routines such as the parameterization of model boundaries, sediment grain size, and bank erosion. Fig. 1 shows a flowchart of model operation along with
required/optional inputs and outputs; these components are discussed throughout this section.

[INSERT FIGURE 1 NEAR HERE]

**2.1 Hydraulics**

The model's hydraulic component is driven using the freely-available, open-source Delft3D software (Version 4.00.01). We employed the model in depth-averaged form (i.e., two-dimensional) as this provided an ideal compromise between computational efficiency and the ability to resolve hydraulics at the cellular scale of our DEMs (Lane et al., 1999). Although Delft3D three-dimensional flow simulations have been applied in river
environments (e.g. Parsapour-Moghaddam and Rennie, 2017), the parameterization, validation, and computational overhead associated with three-dimensional modelling precludes their use for the development and assessment of

MoRPHED (Lane et al., 1999; Brasington and Richards, 2007). Delft3D solves the shallow-water form of the Reynolds-Averaged Navier-Stokes equations, which relate changes in momentum (left-hand terms) in time and space to the cumulative surface and body forces acting on the fluid (right-hand terms):

$$\rho\left(u\frac{\partial u}{\partial x} + v\frac{\partial u}{\partial y}\right) = -\frac{\partial P}{\partial x} + \mu\left(\frac{\partial^2 u}{\partial x^2} + \frac{\partial^2 u}{\partial y^2}\right) \quad [1]$$

$$\rho\left(u\frac{\partial v}{\partial x} + v\frac{\partial v}{\partial y}\right) = -\frac{\partial P}{\partial y} + \mu\left(\frac{\partial^2 v}{\partial x^2} + \frac{\partial^2 v}{\partial y^2}\right) \quad [2]$$

where $x$ and $y$, respectively, denote the streamwise and cross-stream directions of velocity ($u$, $v$). In Equations 1-2, $P$ denotes pressure forces on a body of fluid, $\rho$ is fluid density, and $\mu$ denotes dynamic viscosity. For all modelling, we employed fixed Cartesian orthogonal grids which were generated using the RGFGRID module of the Delft3D software suite. Elevation models used as Delft3D inputs were rotated such that flow was from left (upstream) to right (downstream) for use with a Cartesian coordinate system. The model time step was then adjusted to satisfy the Courant-Friedrichs-Levy condition to ensure computational stability of the solution.

For all simulations, discharge was specified at the upstream boundary and a corresponding water surface elevation was set at the downstream boundary, and was calculated using a normal depth approximation based on reach-average slope and roughness. Horizontal eddy viscosity ($v$) was set at 0.1 s/m$^2$ (Williams et al., 2013). A spatially constant bed roughness was used, based on the Colebrook-White equation to determine the 2D Chezy coefficient:

$$C_{2D} = 18log_{10}\frac{12H}{k_s} \quad [3]$$

where H is water depth and $k_s$ is the Nikuradse roughness length, which can be described in terms of a factor ($\alpha_x$) of the characteristic grain diameter as:

$$k_s = \alpha_x D_{84} \quad [4]$$

Here, we used $D_{84}$ as the characteristic grain size as it provides an estimate of coarse grain influence on the flow field. Using grain size distribution available for both of the modelling sites in this paper (Hodge et al., 2009; Williams et al., 2013), we computed $k_s$ using $\alpha_x$ of 2.9, taken as the average value from a range of gravel-bed rivers

studied in Garcia (2006). This resulted in values of $k_s$ = 0.1 m and 0.29 m for the Rees River and River Feshie respectively.

Models were run with a steady upstream discharge, corresponding to the peak of a flood and hydraulic predictions were extracted for further sediment transport calculations only once the reach had achieved steady state (no observed change in depth, velocity, or inundation). From each hydraulic simulation we exported: (a) water depth, (b) flow velocity resolved into streamwise and lateral components, and (c) bed shear stress. Put simply, flow hydraulics were computed for the maximum discharge of a given flood, and these hydraulics were used to drive subsequent sediment mobilization and morphodynamic evolution. This approach was adopted because: (a) the calculation of morphodynamics once per event allows for greatly reduced computational overhead associated with the model, and (b) modelling at finer timescales, while allowing the ability to capture rapid transient events such as prograding bedload sheets and bank retreat during the course of a single flood, is inherently difficult given that the most common observational data available to geomorphologists describe channel form before and after a single event (Bertoldi et al., 2010; Williams et al., 2011; Mueller et al., 2014), and sediment transport distances, or path lengths, resulting from that event (Pyrce and Ashmore, 2003a,b; Snyder et al., 2009; Kasprak et al., 2015). We acknowledge that a single-flood timestep may necessarily oversimplify morphodynamic evolution resulting from floods that are of sufficient duration to produce multiple entrainment episodes for particles, or those floods that fundamentally alter the bar spacing, and thus the path length distribution, for a reach of interest. While our motivation herein was to assess the validity of a purely event-based model timestep, regardless of flood duration, in Sect. 4.2.1, we provide a nested experiment in which the hydrograph of a single flood event was schematized as a set of three steady discharges corresponding to the rising limb, peak, and falling limb of the event. This schematization was used in an attempt to evaluate (a) the model's suitability for simulating floods of extended duration and (b) the need for quasi-dynamic flow predictions to capture distinct stage-specific morphodynamic processes, such as the dissection of bar-top chutes at falling stages (e.g. Wheaton et al., 2013).

## 2.2 Bed Sediment Erosion

The model employs a critical nondimensional value of the bed shear stress (Shields stress) to determine whether sediment can be entrained at a particular location. The theory and threshold values of Shields stress for entrainment have been well studied in gravel-bed river settings. Incipient motion for gravel occurs when the Shields stress ($\tau_*$) exceeds 0.03-0.07 (Buffington and Montgomery, 1997; Snyder et al., 2009):

$$\tau_* = \frac{\tau_B}{(\rho_s - \rho)gD} \quad [5]$$

where $\rho_s$ is a characteristic sediment density (2650 kg/m$^3$), g is acceleration due to gravity, and $D$ is the median particle size. The spatial distribution of local bed shear stress ($\tau_B$) at steady flow was computed using Delft3D and the critical Shields stress for sediment mobility was set at 0.05. The modelled bed shear varies significantly over small spatial regions (Wilcock et al., 2009) and could therefore lead to large cell-to-cell variability in elevation change and unstable coupling of the bed topography and modelled hydraulics. To avoid such effects, the local $\tau_B$ was computed by averaging the 10 cells longitudinally upstream and downstream of the cell in question along streamlines derived from the Delft3D velocity vectors. Although lateral averaging of shear stress could additionally reinforce model stability, this was not explored here. The result is a single average value of bed shear stress that is used for computation of scour depth at each cell.

Most morphodynamic models compute bed elevation change using some form of the Exner equation for sediment continuity (Paola and Voller, 2005). In this, bed elevation ($z$) through time ($t$) is a function of the sediment supplied from upstream ($V_s$), the divergence of the sediment flux through the reach boundaries ($\nabla Q_s$) and the porosity of the deposited sediment ($\gamma_p$).

$$\frac{\partial z}{\partial t} = \frac{1}{1-\gamma_p}\left(\frac{\partial V_s}{\partial t} + \nabla \cdot Q_s\right) \quad [6]$$

To ensure computational stability, morphodynamics are typically computed by solving Eq. (6) at fine time steps (seconds-minutes) using solutions from the equations of motion. However, MoRPHED is designed specifically to operate at the mesoscale; at model timesteps equal to flood events, (e.g., hours to days), the use of Eq. (6) would lead to the formation of large depressions or mounds in the channel topography, and would drive subsequent computational instability. In short, an event-based model requires an analogous event-based approach for the estimation of bed elevation change. To drive the erosional component of bed elevation change (sediment transport and deposition are discussed in subsequent sections), we used an alternative approach to the local sediment continuity equation, based on Montgomery et al. (1996), who derived a theoretical relationship to predict event-scale sediment scour depth ($D_s$):

$$D_s = \frac{Q_b}{u_b \rho_s (1-\gamma)} \quad [7]$$

where $Q_b$ is the mass-based bedload transport rate per unit channel width during the event, $u_b$ is the bedload velocity, and $\gamma$ is the bed sediment porosity. We note that $Q_b$ was computed here for the peak discharge of any modelled flood; a potential alternative, not explored here, would be to compute $Q_b$ for the average event discharge over the duration of a competent flow. While estimates of $Q_b$ present a continuing challenge, here we use a simple exponential function that relates bedload transport to excess bed shear stress, the latter being readily obtained from the Delft3D simulations.

$$Q_b = (\tau_B - \tau_{BC})^{1.5} \quad [8]$$

In Equation 8, $\tau_{BC}$ is computed from Eq. 5, and $u_b$, the bedload velocity, can be estimated as:

$$u_b = a(u_* - u_{*c}) \quad [9]$$

where $u_*$, the shear velocity, is computed as:

$$u_* = \sqrt{\frac{\tau_b}{\rho}} \quad [10]$$

The constant $a$ in Eq. (9) has been studied by many researchers (Garcia, 2006), but is generally accepted to take a value close to 9. Although the accuracy of using Eq. (7) to predict event-scale scour depth has not been explored empirically, it represents one of the few methods for predicting the depth of scour over extended durations and is intended to be used here in a fundamentally exploratory manner. As alternatives to Eq. 7 become available to geomorphologists in the future, they can be readily substituted into the model.

To summarise, in lieu of a traditional continuity-based approach for estimating bed elevation change (Eq. 6), MoRPHED estimates bed sediment scour depth using an event-based approach (Eq. 7). This scour depth, in combination with the model grid (i.e., cell) size provides a volume of scoured sediment at any location within the model domain. This volume of entrained sediment is transported downstream along hydraulic flowlines and

deposited according to a path-length distribution, each component of which is discussed in the subsections that follow. This routine was performed once per flood, regardless of flood duration; as our intent was to develop a purely event-scale morphodynamic model, we did not attempt to scale or correct estimates of bed sediment erosion as a function of flood duration in this study.

## 2.3. Bank Sediment Erosion

Readily-erodible banks are a hallmark of braided rivers (Wheaton et al., 2013) yet continue to represent one the most difficult geomorphic processes to represent numerically (Darby and Thorne, 1996; Simon et al., 2000; Rinaldi and Darby, 2007; Stecca et al., 2017). Most existing models rely on fine timesteps and detailed predictions of the near bank force balance to predict bank stability. As MoRPHED is a simplified event-scale model, here we estimate the lateral retreat distance by empirically scaling the distance of lateral retreat during a model run to near-bank shear stress and bank slope.

To begin, the model calculates the slope of all cells in the model domain; all simulations presented herein used a cell resolution of 2 m (Rees River; Sect. 3.1) or 1 m (River Feshie; Sect. 3.2). The relatively coarser cell resolution on the Rees was chosen for computational efficiency, particularly with regard to the hydraulic modelling component of MoRPHED, as the Rees site encompassed a considerably larger spatial domain than that of the Feshie. Cells that exceed a user-defined slope criterion, which was set at 7% by examining the average slope of cells that underwent bank retreat in field surveys, for all simulations presented herein (Fig. 2.1), are then identified as candidate cells for undergoing bank erosion. Whether bank erosion occurs by mass failure or lateral channel migration, eroding banks are often marked by steep slopes, and as such this criterion was used as the first metric for computing bank sediment erosion. While below the angle of repose for both consolidated and unconsolidated sediment, this 7% threshold simply served as a first approximation of those areas that may exhibit sufficient slope to undergo bank retreat and was designed to be inclusive of areas that might be below slope failure criteria, but may nevertheless undergo bank retreat as a function of near-bank shear stress, discussed below. This slope delineation produces groups of cells, which are removed from the selection if the group's area falls below a user-specified threshold (Fig. 2.2). This area cutoff was used for computational efficiency, and was here set to 30 cells (i.e., 30 m$^2$ on the Feshie and 120 m$^2$ on the Rees), which produced good agreement between model-predicted bank erosion and field-observed bank erosion patches; we also observed that very small area thresholds would create discontinuous patches of lateral retreat, leading to model instability and hydraulic artefacts in subsequent runs.

For each cell within these groups, the bed shear stress in the surrounding cells was sampled using a 3 x 5 neighbourhood window oriented in the cardinal direction of the candidate cell's aspect. The bed shear stress values in these cells was then averaged, producing a single shear stress value for the delineated group of cells. Those cell groups with average shear stress below a user-defined criteria, here set to 50 N/m$^2$, were excluded. The use of near-bank shear stress as a predictor variable for estimating lateral bank retreat has been applied by numerous researchers, including Ikeda et al. (1981), Howard (1992; 1996), Sun et al., (1996; 2001), and Stølum (2001).

The cells that remained were those exceeding both the slope and near-bank shear stress thresholds, and for each of these cells, the model then removed material from a number of adjacent cells specified by Eq. (11) and shown in Fig. 2, panel 4A, which computes the lateral extent of bank erosion (n), rounded to the nearest whole cell, shown in Fig. 2, panel 4, as a function of slope (S) and near-bank shear stress (τ; Fig. 2.4):

$$n = round((\frac{\tau}{3} + 1) * \frac{S}{15}) \quad [11]$$

The location of these *n* cells was determined by moving away from the initial candidate cell at one-cell increments, in a direction opposite the initial candidate cell's cardinal aspect direction (i.e., simulating lateral bank retreat moving away from the channel). The number of cells adjacent to the initial candidate cell is shown in Fig. 2.4. All delineated cells were then surrounded by a 3 x 3 neighbourhood window, and the full group of candidate cells, at the conclusion of this delineation routine is shown by the red polygon in Fig. 2.4A. Within each of these 3 x 3 neighbourhoods, all cells were reduced in elevation to a level equal to that of the lowest cell in the neighbourhood (sensu Nicholas, 2013), and eroded sediment was immediately transported downstream and deposited in a manner identical to that described below for bed transport and deposition.

Bank erosion and bank material transport/deposition are computed prior to bed morphodynamics (Fig. 1), and to conserve computational overhead, hydraulics are not recomputed between these steps; as a result, bed scour is not altered in areas proximal to eroding banks. Further, because candidate cells for bank erosion are identified simply on the basis of slope and shear stress, it is possible that cells away from the wet/dry boundary (i.e., the channel bed) can undergo erosion in this manner as well. We believe this approach is appropriate here, as hydraulics and morphodynamics are computed only once per flood (at peak discharge), and as such, steeply-sloping cells

susceptible to erosion may be inundated within the channel at the event peak, but may still undergo subaerial failure on the rising or falling limb of the hydrograph. We note that Equation 11 was used for simulations on both river systems studied here, which varied considerably in their constituent volumes of fine/cohesive sediment and vegetation extent (see Section 3 below). Thus, these values are likely applicable for a variety of rivers, but we caution that for systems with exceptional bank cohesion, whether via vegetation or fine sediment, adjustment of these parameters may be necessary. Finally, as opposed to a physically-based relationship between shear stress, bank slope, and the extent of lateral erosion, Equation 11 was derived through qualitative calibration; that is, rather than presenting a deterministic methodology for quantifying bank retreat, this approach is simply reflective of the best qualitative correspondence between modelling results and field observations of areas undergoing bank erosion.

[INSERT FIGURE 2 NEAR HERE]

## 2.4 Bed/Bank Sediment Transport and Deposition

Once entrained, bed or bank sediment is mobilized downstream along flowlines which are delineated using velocity vectors from Delft3D. Although MoRPHED is inherently a cell-based (i.e. raster) model, Delft3D-derived velocity vectors did not necessarily pass through the centre of any given cell within the MoRPHED model grid. To account for this, the nearest grid cell was computed along the Delft3D velocity vectors at downstream intervals equal to the computational domain cell size (i.e., 1 m on the Feshie, 2 m on the Rees; see Section 3). In the field, deposition of sediment consistently occurs in diffuse patterns, such as 'tear-drop' forms of lobate bars, prograding bedload sheets, and thinly-mantled overbank deposits (Ashmore, 1982; Ferguson and Werritty, 1983; Wheaton et al., 2013). To mirror this diffuse deposition, the model distributes sediment within a 5x5 window of cells surrounding the candidate deposition cell, with the candidate cell receiving 1/3 of the deposited sediment, the adjacent eight cells receiving 1/3 of the deposited sediment between them, and the outer ring of 16 cells receiving the final 1/3 of deposited sediment. In the case of dry cells that occur in the 5x5 neighbourhood, the dry cell(s') sediment is divided among the population of wetted cells in the neighbourhood. The total volume of deposited sediment within this 5x5 window is equal to the fraction of all entrained sediment from upstream given by a user-defined path-length distribution at the centre cell's distance downstream from the entrainment location (Fig. 3).

At each cell along the flowpath, the volume of sediment to deposit in the centre cell is given by a path length distribution (Fig. 4). In the simplest sense, this distribution details the proportion of all eroded sediment which is

deposited at a particular distance downstream, and provide a representation of field-mapped source-sink pathways found in braided rivers (Williams et al., 2015). These distributions have been studied by numerous researchers and found to take several forms in braided rivers, as reviewed in Sect. 1. MoRPHED deposits a proportion of the path-length specified volume of sediment in each wetted cell of the 5x5 neighbourhood described above. Path-length

distributions in the model can take typical field-measured forms (Gaussian and Exponential Decay; Pyrce and Ashmore, 2003b), or can take any user-specified form (e.g. multi-peaked), as specified by an input text file. Because sediment deposition is simply computed via a provided path length distribution (and particles are not dynamically tracked on their journey downstream), sediment transport effectively occurs instantaneously in the model. As such, a key assumption within MoRPHED is that modelled events are of sufficient duration to allow particles to transit

the full length of the specified path length distribution. Laboratory results of gravel-bed braided rivers have, however, indicated that particle transport and deposition, and development of a path length distribution, occurs quite rapidly after the onset of competent flow (see Kasprak et al., 2015). At any given cell within the model domain, elevation change is computed as the difference between erosion and deposition at that location. It is possible for any particular cell, over the course of a model run, to experience erosion exclusively, deposition

exclusively, both erosion and deposition, or a total lack of sediment scour and deposition.

[INSERT FIGURE 3 NEAR HERE]

**2.5 Sediment Import and Export**

For each simulated event, the model tracks the volume of sediment passing the downstream or lateral reach boundaries. In effect, export of sediment occurs when the length of the user-specified path length distribution exceeds the downstream or lateral boundaries of the model domain. When this occurs, the remaining volume of sediment (the amount of eroded sediment not yet deposited along the flowline) is recorded as having been exported

from the reach. Sediment import is user-specified and can be (a) set equal to the volume of sediment export during the preceding event (e.g. sediment equilibrium; Grams and Schmidt, 2005), (b) specified as a percent of sediment export during the preceding event, or (c) specified via a text file detailing absolute volumetric sediment import during each event (e.g. sedigraph timeseries). Algorithmically, the model computes flowpaths from each wetted cell at the upstream reach boundary and distributes the total volume of imported sediment to each cell of each

flowpath as specified in the user-input path length distribution. Sediment is introduced to the reach at the end of each event (Fig. 1; i.e., following hydraulic and bed/bank morphodynamic modelling) and prior to initiation of the

subsequent event, as this allows for the import of sediment once exported sediment volumes are known. Sediment leaving the lateral reach boundaries was included in the total export volume, along with sediment leaving via the downstream reach boundary.

## 2.6 Model Verification

To compare the outputs of the model with field-based surveys of channel evolution, we derived several morphometric parameters along with comparing DEMs-of-Difference (DoDs) and contributions of individual
braiding mechanisms to total geomorphic change. Each of these three verification components are discussed below.

### 2.6.1 Morphometric Indices

We manually quantified braiding index ($I_B$) for the initial and final field surveys and model runs of each simulation
described here. Braiding index was computed by averaging the number of channels across five evenly spaced transects along the length of the model domain (Howard et al., 1970; Egozi and Ashmore, 2008). Channels were defined by wetted areas as modelled using Delft3D at estimated baseflow for each modelling site. In addition, we measured the total sinuosity ($S_T$) of the modelled reach for the first and last model runs in each system. Total sinuosity (Richards, 1982) was defined by the ratio of the length of all anabranches ($L_A$) compared to the down-
valley length of the model domain ($L_D$):

$$S_T = \frac{L_A}{L_D} \quad [15]$$

Finally, we computed the number of confluences, diffluences, and channel heads for initial and final field and
model DEMs. The procedure for delineating confluences, diffluences, and channel heads is detailed by Wheaton et al. (2013). In brief, it requires manual location of areas where one anabranch splits into two anabranches (a diffluence), areas where two anabranches join to form one anabranch (a confluence), and locations where small side channels or chutes begin (a channel head). In theory, the number of confluences should be roughly equal to the number of diffluences plus the number of channel heads for a braided river. Instances where diffluences and
channel heads outnumber confluences are indicative of distributary systems (e.g. deltas; Jerolmack and Morhig,

2007), while sites dominated by confluences are indicative of dendritic networks that collect low order flow paths into a few main channels.

### 2.6.2 DEMs-of-Difference

We differenced DEMs of initial and final field surveys and model simulations using Geomorphic Change Detection software (version 6, http://gcd.joewheaton.org; Wheaton et al., 2010). Differencing DEMs from two survey periods produces a DEM-of-Difference (DoD), or a map of geomorphic changes that occurred during the inter-survey period. While the DEM differencing process is straightforward, accounting for error in the resultant DoD is

necessary as each constituent DEM contains an inherent level of error (that may vary on a cell-by-cell basis), which can ultimately influence the estimated magnitude of geomorphic change in the DoD. Here we modelled DEM error using the most straightforward of the available approaches: we simply assumed that each of the constituent DEMs contained no error and computed the DoD. We then thresholded, or removed areas of change, from the DoD if they were less than ±0.1 m in magnitude (herein termed the 'minimum level of detection' or minLoD). For decadal-scale

modelling and field surveys on the River Feshie, we employed a threshold of ±0.2 m to better visualize and delineate braiding mechanisms and to account for the larger magnitude elevation changes that occurred over wide swaths of the braidplain over this extended timescale. While this simple thresholding method is not necessarily the most robust available for modelling error in field-surveyed DEMs, those DEMs output from the model do not contain survey error as would be expected from field-based DEMs. As such, a simple minLoD of 0.10 m allowed us to use

identical error modelling methods to directly compare areas of change in field and modelled DoDs while simultaneously removing a great deal of the survey noise/uncertainty present in field DEMs, along with removing extremely low-magnitude elevation changes in modelled DEMs.

From DoDs, we extracted elevation change distributions (a histogram of all volumetric changes), along with

deriving the net sediment imbalance during each survey and model period (the percent departure of the sediment budget from equilibrium conditions).

### 2.6.3 Braiding Mechanisms

Using GCD software, we mechanistically segregated both field and model-derived DoDs by delineating the processes responsible for each area of geomorphic change (Ashmore, 1991; Wheaton et al., 2013). These processes

can be separated into four 'braiding mechanisms' described by Ashmore (1991): central bar development, lobe dissection, transverse bar dissection, and chute cutoff. To these, Wheaton et al. (2013) added an additional six mechanisms that are not unique to braided rivers: bank erosion, channel incision (i.e. bed erosion), overbank sheets, confluence pool scour, bar trimming, and lateral bar development. Hereafter, these ten processes are referred to simply as 'braiding mechanisms'. We mapped the areas in DoDs where each of these mechanisms occurred; areas where we could not confidently assign a mechanism of change were classified as questionable or unresolved change. While this approach is inherently subjective, the delineation of braiding mechanisms, and more broadly mechanisms of geomorphic change, has been completed for numerous study sites (Wheaton et al., 2013; Kasprak et al., 2015; Sankey et al., 2018a,b), and efforts aimed at the automated mechanistic segregation of geomorphic change have also been presented in the literature (Kasprak et al., 2017). This classification allowed us to compute and compare the volumetric contribution of each braiding mechanism to total geomorphic change in both field and model-derived DoDs.

## 3 Study Sites

### 3.1 Rees River, South Island, New Zealand - Event and Annual Scales

The braided gravel bedded Rees (Fig. 5) drains a 402 km$^2$ catchment of the uplifting metasedimentary Southern Alps and flows into Lake Wakitipu. The 2.5 km study reach is an actively braided channel which flows through a deglaciated valley, and the river is braiding in response to sediment delivery from the tectonically-active landscape (Williams et al., 2013). The hydrology of the system is dominated by response to both seasonal snowmelt and rainfall, and undergoes floods in the spring, summer, and fall that may completely alter the morphology of the braidplain over the course of a single flood. A temporary gauging station at Invincible (6 km upstream of the study reach) operated from 2009-2011 and is used to drive hydraulic components of the model. Mean discharge during the 2010 and 2011 hydrological years was 20 m$^3$s$^{-1}$, with a maximum flow of 475 m$^3$s$^{-1}$ (Williams et al., 2015). The survey data on the Rees include 0.5 m resolution digital elevation models (DEMs) constructed via a fusion of terrestrial laser scanning (TLS) and optical-empirical bathymetric mapping surveys (Williams et al., 2014), which were downsampled to 2 m resolution for modelling. In total, ten floods ranging from 51 m$^3$s$^{-1}$ to 403 m$^3$s$^{-1}$ were captured as part of the ReesScan project (Brasington et al., 2012) from 2009-2011, with post-flood DEMs surveyed in the period between each high flow. These pre- and post-flood DEMs, along with a continuous hydrologic record

and a high degree of dynamism across the braidplain at the event scale, make the Rees an ideal candidate to examine the performance of the model at the event and annual scales.

## 3.2 River Feshie, Scotland - Annual and Decadal Scales

The weakly-braided gravel bedded Feshie (Fig. 5) is a tributary of the Spey River and drains 231 km$^2$ of mountainous, postglacial terrain. Underlain by metamorphic and igneous rocks, the basin ranges from 230 m - 1260 m in elevation. The mean flow near the river's outlet was reported by Ferguson and Werritty (1983) as 8 m$^3$s$^{-1}$ with $Q_5$ = 80 m$^3$s$^{-1}$. Topographic data for the 1 km study reach of the Feshie consist of nine years of resurveys (2000,
2002-2008, 2013) comprising more than a decade of channel change using RTK-GPS (2000-2006) along with TLS and RTK-GPS fusion scans performed for three surveys (2007-8, 2013). DEMs generated from field surveys on the Feshie had a cell resolution of 1 m. Additionally, the Feshie dataset contains continuous hydrograph data (~55 years) and aerial photo records (~60 years), along with UK Ordnance Survey channel planform maps dating to 1869. The Feshie has been the site of a great deal of previous research ranging from bar morphodynamics (Ferguson
and Werritty, 1983; Wheaton et al., 2013) to development of riverine survey and DEM-differencing/change detection methodologies (Brasington et al., 2007; Vericat et al., 2007; Hodge et al., 2009; Wheaton et al., 2010). The combination of annual resurveys capturing over a decade of channel change in combination with mapping and aerial photographs dating back over a century make the Feshie an ideal candidate to examine the performance of the model at the annual and  decadal scales. In addition, the Feshie site provides a mechanistic contrast to the Rees
in that overall flood-to-flood dynamism is reduced via vegetation cohesion and fine sediment (Ferguson and Werritty, 1983), whereas the comparatively finer and more labile Rees undergoes widespread geomorphic change across the braidplain, inhibiting the establishment of vegetation. On the Feshie, the dominant mechanisms of change vary from those seen on the Rees, particularly with regard to chute cutoff and bank erosion (Wheaton et al., 2013).

[INSERT FIGURE 5 NEAR HERE]

# 4 Results

## 4.1 Hydraulic Model Calibration

### 4.1.1 Rees River

Because we modelled the same reach of the Rees previously analysed by Williams et al. (2013), we simply used the values of $k_s$ (0.10) and $\upsilon$ (0.10) obtained in that study (Table 1). We found that these values resulted in good agreement between field-measured and modelled water depth, velocity, and inundation extents. The reader is referred to Williams et al. (2013) for more detailed analysis of the validity of Delft3D on the Rees (and in braided gravel-bed rivers in general).

### 4.1.2 River Feshie

In contrast to the Rees, no comprehensive validation and verification of Delft3D exists on the Feshie; as such, we leveraged existing surveys of wetted areas from 2003-2007 in concert with surveyed water depth in those years to examine the performance of Delft3D. Because field surveys were conducted at low flows to facilitate rapid measurement of braidplain topography, here we are only able to verify the results of Delft3D at these low flows. However, Delft3D has been employed and verified on gravel-bed braided rivers at flood stage (Javernick, 2013), demonstrating that the model can accurately reproduce flood-stage hydraulic features and can be used to drive morphodynamic evolution at the event-scale. For modelling on the Feshie, we estimated discharge by downscaling the average observed flow for the relevant survey period at the nearest gauging station (SEPA No. 8013, Feshie at Feshiebridge) located approximately 11 km downstream, using an empirically-derived discharge-area coefficient of 0.71 (Wheaton et al., 2013). The value of discharge taken at Feshiebridge was the average flow during each year's survey period (average of two weeks). We estimated the downstream water elevation using surveyed inundation extent in combination with the DEM for each year modelled. Downstream water surface elevation estimated from the spatial data were cross-checked using a reach-scale conveyance calculation (Williams et al., 2013).

Results of our validation of Delft3D on the Feshie at low flow are shown in Fig. 6. Here we report (a) the mean of depth differences between modelled and observed values ($D_{diff}$), along with (b) the congruence of the modelled and

measured inundation extents ($F_c$; Bates and De Roo, 2000) as described by the ratio of intersection and union areal extents. These two metrics are described by Eqs. (16-17), respectively.

$$D_{diff} = \frac{\sum_{i=1}^{n} x_{mod} - x_{obs}}{n} \quad [16]$$

$$F_C = \frac{IA_{obs} \cap IA_{mod}}{IA_{obs} \cup IA_{mod}} * 100 \quad [17]$$

The metrics indicate that at low flow, Delft3D accurately predicted both depth and inundation extent across the Feshie study reach. Both $D_{diff}$ and $F_c$ are consistent with values obtained by Williams et al. (2013) on the Rees (mean $D_{diff}$ = 0.04, mean $F_c$= 81.2 on the Feshie) and are indicative of good agreement between hydraulic predictions and field-observed flow characteristics, despite the empirically scaled discharge.

[INSERT FIGURE 6 NEAR HERE]

## 4.2 Event-Scale Morphodynamic Modelling: Rees River

We modelled a single flood event on the Rees River that occurred between 8-16 December 2009 as the result of heavy rainfall in the upstream watershed (Fig. 7). Peak flows reached a maximum instantaneous discharge of 259 m$^3$s$^{-1}$ at the upstream Invincible gauging station during the afternoon of 9 December. Two smaller peaks in the hydrograph of 75 m$^3$s$^{-1}$ (afternoon of 8 December and morning of 12 December) also occurred during this event. Our modelling employed a single representative grain size ($D_{50}$) of 20 mm. We used an equilibrium sediment budget condition for this simulation (Sect. 2.5). Geomorphic change captured by pre- and post-flood terrestrial laser scanning revealed that the most volumetrically significant mechanisms of change were transverse bar conversion (22%), bank erosion (21%), and lobe dissection (19%). Qualitatively, event-scale dynamics across the study reach are marked by widespread geomorphic change, particularly in the centre of the braidplain where development of a single main channel occurred via channel incision, bank erosion, and avulsions of smaller anabranches leading to deposition and subsequent dissection of mid-channel bars. Geomorphic change on the edges of the braidplain was somewhat muted, consisting largely of infilling of anabranches and accretion of central bars. Field-surveyed elevation changes ranged from -1.70 m to +1.32 m (Fig. 7D). The braiding index ($I_B$) decreased from 3.6 to 2.0 following the flood, and total sinuosity ($S_T$) decreased from 4.5 to 2.9.

[INSERT FIGURE 7 NEAR HERE]

Results of morphodynamic modelling on the Rees for this event are shown in Fig. 7. This event-scale model was run using a steady-state discharge of 259 $m^3s^{-1}$ as seen in the December 2009 flood peak. Total model runtime for the single event was approximately 90 minutes. Modelled elevation changes in the study reach ranged from -1.76 m to +0.89 m (Fig. 7E); these, and subsequent modelling results incorporate a 0.1 m minimum level of change detection (i.e., threshold), as discussed in Section 2.6. Overall, geomorphic change was concentrated near the centre of the braidplain, similar to geomorphic change measured from field data. Large swaths of bank erosion along a central anabranch developed, although not to the extent seen in field data. In general, geomorphic change in the modelled DoD appears muted in comparison to the field-based DoD. This is reflected in the elevation change distribution (ECD) shown for field and model data on the Rees (Fig. 7C), particularly with regard to the erosional component of volumetric change (47,598 $m^3$ in the field compared to 20,663 $m^3$ in the model; Table 2). Similarly, depositional volumes were greater in the field (35,551 $m^3$) compared to those in the model (16,188 $m^3$; Table 2). On average, erosion depth across the study reach was 0.13 m as observed through field measurement, and 0.07 m when modelled. Deposition averaged 0.10 m in the field and 0.06 m in the model (Table 2). The most volumetrically-significant braiding mechanisms in this run were central bar development (25% of total volumetric change), bar edge trimming (16%), and bank erosion (15%).

[INSERT TABLE 2 NEAR HERE]

### 4.2.1 Case Study: Hydrograph Discretization

The choice of model timestep is one of the more important considerations in morphodynamic modelling, whereby the user must strike the optimal balance between a model timestep fine enough to preserve computational stability, yet which is coarse enough to allow computation over meaningful timescales (Brasington et al., 2007). To investigate these questions using MoRPHED, we discretized the hydrograph used in event-scale modelling on the Rees so as to model three discrete points over the course of the modelled flood (Fig. 8). These discharges were 75, 259, and 75 $m^3s^{-1}$, respectively. Note that in this case study, we did not alter the timestep of the model itself, which remained at the event scale, but rather computed morphodynamic change at three instances over the course of a multi-day flood with three sub-peaks, using an equilibrium sediment budget. Planform DoD, ECD, and volumetric contribution of individual braiding mechanisms from this discretized model run are shown in Fig. 8. In general,

morphologic changes were more widespread across the braidplain in the case of the discretized hydrograph modelling run than the single peak discharge hydrograph (Fig. 8), with overall area of change increasing, yet still smaller than field-observed change volumes (157,504 m$^2$ in the model compared to 287,184 m$^2$ in field data; Table 2). The ECD from this model run more closely approximated the field-derived ECD, with low-magnitude elevation changes (e.g. < 1 m) dominating the change distribution (Fig. 8).

[INSERT FIGURE 8 NEAR HERE]

## 4.3 Annual-Scale Morphodynamic Modelling: River Feshie

We modelled morphodynamics during the one-year period from July 2003 to July 2004 on the Feshie. The estimated hydrograph for the study reach, based on the empirical downscaling coefficient applied to the Feshiebridge gauge (Sect. 3.2) during this period is shown in Fig. 9A. This survey epoch contained 16 flood peaks above competence (20 m$^3$s$^{-1}$; estimated by Ferguson and Werritty, 1983) at the study reach; these are denoted in Fig. 9 and were used as model inputs. Ashworth and Ferguson (1989) subsequently documented that flows of 20 m$^3$s$^{-1}$ were indeed competent for bed material in the study reach, albeit without full mobility of all bed particle sizes. Our modelling on the Feshie employed a single representative grain size ($D_{50}$) of 50 mm.

[INSERT FIGURE 9 NEAR HERE]

Wheaton et al. (2013) noted that the most volumetrically significant braiding mechanisms during the 2003-2004 period were chute cutoff (29%), bank erosion (16%), and channel incision (15%). Overall geomorphic change was primarily confined to a main channel bisecting the braidplain longitudinally, with one anabranch on the left side of the braidplain undergoing bank erosion and central bar development. Braiding index ($I_B$) during the 2003-2004 epoch increased from 1.8 to 2.8, and total sinuosity ($S_T$) also increased from 2.5 to 3.5. Results of morphodynamic modelling are shown in Fig. 9E. Total model runtime for the series of 12 events modelled during the 2003-2004 period was approximately 6 hours. The results of DEM differencing are shown in Table 3. Elevation changes in the modelled reach ranged from -1.25 m to +1.49 m. Overall geomorphic change was marked by the accumulation of transverse bars and the development of central bars throughout the model reach, along with incision of a central anabranch. Sculpting or trimming of central bars (Fig. 9F; e.g. Wheaton et al., 2013) was also prevalent. The most volumetrically-significant braiding mechanisms during this model run were channel incision (26%), transverse bar

conversion (20%), and central bar development (14%). Overall, geomorphic change predicted via modelling was greater than that observed from field data (Table 3). However, both field and model ECDs (Figs. 9B, 9C) depict change distributions wherein the greatest volume of geomorphic change is the result of low-magnitude elevation changes. The average depth of elevation changes were generally well predicted by the model, although average

erosion and deposition depths were over-estimated by 0.02 and 0.03 m, respectively (Table 3).

[INSERT TABLE 3 NEAR HERE]

### 4.3.1 Case Study: Contrasting Path-Length Distributions

Transport and deposition of eroded bed or bank sediment in MoRPHED is a function of the path length distribution used in the model. While field or laboratory data describing particle transport distances can be used to produce a path length distribution, parameterizing such a distribution for sites where tracer data are not available is not straightforward. Here we employed average confluence-diffluence spacing to estimate the peak of the distribution

(Fig. 3; Pyrce and Ashmore, 2003a,b; Kasprak et al., 2015). To further understand the effect of contrasting path length distribution shapes and distances, we used MoRPHED to model the annual hydrograph on the River Feshie in a manner identical to Sect. 4.3, except we varied the characteristics of the specified path length distribution (Fig. 10). We employed a compressed Gaussian distribution (Fig. 10A), a flattened Gaussian distribution (Fig. 10B), and an exponential decay-type distribution (Fig. 10C; Pyrce and Ashmore, 2003b); each of these distributions had a

total length identical to our original distribution on the Feshie (Fig. 10A). We also modelled two shortened distributions (length = 50 m): a shortened Gaussian distribution (Fig. 10D) and a shortened exponential distribution (Fig. 10E). As previously stated, we assume that event duration is sufficient for particles to transit the full length of the specified path length distribution, and in all simulations we employed an equilibrium sediment budget (Sec. 2.5), where the amount of imported sediment was set equal to that amount crossing the downstream and lateral

model boundaries.

[INSERT FIGURE 10 NEAR HERE]

Overall, the geomorphic changes predicted by modelling differing path length distributions were strikingly similar

at the reach scale, and areas of scour and deposition generally aligned between all five of the simulations (Figs. 10A - 10E). Analysis of the elevation change distribution produced using each path length distribution revealed

that while the compressed and stretched Gaussian distributions were marked by more laterally-extensive deposition at higher magnitudes (e.g. ~1 m), the exponential and two shortened distributions (Figs. 10C - 10E) generally contained depositional signatures marked by numerous low-magnitude (e.g. $\Delta z < 0.5$ m) changes. The same is true for the erosional component of elevation change, with compressed and stretched Gaussian distributions marked by a wide range of erosional elevation changes up to and exceeding 1 m, whereas the exponential and shortened distributions generally displayed erosional changes less than -1 m in elevation.

## 4.4 Decadal-Scale Morphodynamic Modelling: River Feshie

We modelled morphodynamics during the ten-year period between July 2003 and June 2013 along the River Feshie. The estimated hydrograph at Glen Feshie during this period is shown in Fig. 11A, and using an equilibrium sediment budget, we modelled all peaks above 20 $m^3s^{-1}$ as described in Sect. 3.2 and in Wheaton et al. (2013), for a total of 185 flood events ranging from 20 $m^3s^{-1}$ to 95 $m^3s^{-1}$.

Differencing DEMs from survey data at the beginning and end of the analysis period reveals elevation changes ranging from –2.4 m to +2.1 m (Fig. 11D). DEM differencing indicates that the study reach underwent slight net aggradation (+3.6% imbalance). The most volumetrically significant braiding mechanisms during this time period were the development of central bars (25% of volumetric changes), transverse bar conversion (17%), and bank erosion (16%). As the relative contribution of individual braiding mechanisms may be misleading at decadal scales due to signature overprinting and hence difficulty in interpretation of braiding mechanism, we note that over a 5-year period (2003-2007; Wheaton et al., 2013) on the Feshie, the most volumetrically significant braiding mechanisms were chute cutoff (24%), bank erosion (20%), and transverse bar conversion (19%). Braiding index ($I_B$) during the 2003-2013 epoch increased from 1.8 to 2.4, and total sinuosity ($S_T$) also increased from 2.5 to 2.9 (Fig. 11).

Results of morphodynamic modelling from 2003-2014 are shown in Fig. 11E. Total model runtime for the 185-flood series was approximately 72 hours. Geomorphic change ranged from +2.1 m to -8.07 m (Fig. 11E). A mask was employed to exclude areas of the reach < 25 m from the upstream boundary, as boundary artefacts resulting from the use of point discharges in the Delft3D model produced high magnitudes of geomorphic change (e.g., > 5 m) in these areas. While geomorphic change in the field was marked by generally thin-mantled erosion and deposition across the braidplain, the model produced more widespread, high-magnitude erosional change (Table

4). While the depositional component of the ECD produced by the model generally characterized that of the field-derived ECD (Figs. 11B, 11C), the model predicted a smaller area, but greater volume, of scour (Table 4). The model generally reproduced the form and magnitudes of deposition seen in the field, but the lowest-magnitude deposition (e.g. < 0.5 m) was more volumetrically significant in the field than in the model. While the model did produce avulsions (for example, the development and incision of a new channel on the left-hand side of the upstream end of the braidplain in Fig. 11), it did not produce avulsions in the same locations seen in the field, nor with the same frequency, as evidenced by the model predicting an incised, simplified channel network in 2013; this downcutting behaviour was also seen by Singh et al. (2017) in decadal-scale morphodynamic modelling using Delft3D. This central anabranch incision was accompanied by a reduction in channel nodes (confluences, diffluences, and channel heads), and thus a simplified channel planform, in the model as compared to the field dataset (Fig. 11). The most volumetrically-significant braiding mechanisms during the 2003-2013 model run were central bar development (27%), transverse bar conversion (22%), and lobe dissection (1%). In addition, the role of bar edge trimming (8%), a process treated identically to bank erosion in MoRPHED, was magnified compared to field-derived mechanistic segregation (2%), as was the role of channel incision (11% in the model as compared to 7% in the field).

[INSERT FIGURE 11 AND TABLE 4 NEAR HERE]

## 5 Discussion

We developed a morphodynamic model that computes sediment transport according to user-specified path length distributions, and subsequently employed this model to predict channel evolution at two braided river reaches across timescales ranging from a single event to a decade. We observed that the model reproduced many of the geomorphic changes observed in the field, although the magnitude and mechanisms of those changes often diverged from observations using field data. At the same time, the modular design of the modelling framework may hold promise for exploration of braided channel evolution, and also raises questions regarding the way processes are represented algorithmically and the model's sensitivity to those process representations.

### 5.1 Emergent versus parameterized processes

Braided rivers undergo geomorphic change as a result of numerous morphodynamic processes, or braiding mechanisms (Ashmore, 1991; Wheaton et al., 2013; Kasprak et al., 2015). Given its highly simplified nature, the degree to which these braiding mechanisms must be explicitly represented as algorithms in morphodynamic modelling deserves exploration. Of the ten braiding mechanisms discussed in Sect. 2.7.3, the model produced eight

simply as a result of the bed erosion, transport, and deposition functions included in the model. Only bank erosion and bar edge trimming required the inclusion of a lateral channel migration component. As the surfaces undergoing bank erosion and bar edge trimming are not necessarily submerged during a flood, these processes cannot be captured simply by an excess shear stress scour approach (Eq. (7); Sect. 2.2).

The geomorphic changes that the model most commonly produces are those that result from focused scour and longitudinally-continuous deposition, given the nature of the scour and deposition functions used in the model. In particular, channel incision, bar edge trimming, bank erosion, and lateral/central bar development are common processes produced by the model (Figs. 7, 9, 11). Additionally, given the single-peaked Gaussian distributions used herein, those braiding mechanisms which involve deposition immediately downstream of an erosional source

are difficult to reproduce. For example, Wheaton et al. (2013) demonstrated the importance of chute cutoff as a braiding mechanism on the Feshie, noting that chute development across point bars not only manifested as erosion, but that the scoured material was often deposited immediately downstream of the chute. Similarly, scoured bank material (e.g. mass failures) may often be deposited at the bank toe rather than transported downstream. In both cases, the model makes no differentiation in transporting the eroded sediment, mobilizing the material according

to the user-specified path-length distribution; as such, proximal couplets of erosion and deposition are difficult to reproduce in the model. Finally, we note that chute cutoff in the model always occurred in locations of pre-existing chutes across point bars. Because chute cutoff often occurs at the falling stage of floods, when braidplain-inundating flows are first being confined into anabranches, our model may not properly reproduce chute cutoff as a result of only computing peak flood hydraulics, and averaging shear stress across a range of model cells. Although headward

erosion of these pre-existing chutes typically occurred in the model, thus increasing their extent, we did not observe any instances where chute cutoff was initiated in the model without an existing chute or channel head being present on a bar surface. As such, chute cutoff may represent a braiding mechanism that must be explicitly included in the model's code in order to be properly represented in the future.

Finally, it is worth noting that we did not explicitly include avulsion as one of the ten braiding mechanisms examined here, for consistency with the mechanisms analysed by Ashmore (1992) and Wheaton et al. (2013), and

also because avulsions, by nature, arise from a combination of the examined braiding mechanisms; that being said, avulsion is certainly essential to the development and maintenance of braided planform (Ferguson, 1993). While the model is algorithmically capable of producing avulsions through a combination of lateral bank retreat and sediment scour/deposition, these did not occur at the same frequency as was seen in the field; this is particularly evident in the decadal-scale simulation's incised and simplified channel planform, along with the reduction in channel nodes as compared to field observations (Fig. 11). At these extended timescales, the model appears to exhibit a positive feedback loop where bed sediment scour drives increased flow capture and bed shear stress, resulting in further downcutting of the channel. Adjustments in bed sediment scour, sediment deposition (in particular, more focused deposition of material to offset erosion and drive flow deflection), along with increased rates of lateral bank erosion all have the potential to drive more frequent avulsions, prevent over-scouring of the bed, and provide greater fidelity to field observations of channel morphodynamics when modelling at decadal and longer timescales.

## 5.2 Sensitivity to process representation

### 5.2.1 Hydrograph discretization

In Sect. 4.2.1, we modelled a single event on the Rees as three discrete discharges on the hydrograph (Fig. 8). Because MoRPHED under-predicted the volume of change, particularly due to the absence of low-magnitude scour, during the single-event simulation (Fig. 7), we sought to understand whether discretizing the hydrograph would allow for an improved prediction of overall volumetric change, and low-magnitude erosional change in particular. Overall, discretizing the hydrograph into three modelling timesteps only marginally increased predictions of volumetric change across the study reach: total volumetric change in the discretized run (Fig. 8) was 39,771 $m^3$, compared to 36,851 $m^3$ in the single-event model run (Fig. 7). Both modelling approaches underestimated the amount of volumetric change in the field (83,149 $m^3$).

Discretizing the hydrograph also increased the amount of low-magnitude scour predicted by the model (Fig. 8B), more accurately reflecting the field-derived elevation change distribution (Fig. 7B). It is likely that this is the result of the 0.1 m elevation thresholding used in our change detection (Sect. 2.7.2), whereby additive changes due to erosion largely did not exceed 0.1 m depth after a single flood event, but did exceed this threshold when three discrete hydrograph points were measured. Whereas erosional processes that lead to high-magnitude scour, such

as bank erosion, dominated the elevation change distribution in the single-event model run, processes such as channel incision and lobe dissection were more prevalent in the discretized hydrograph model run. Additionally, several areas of high-magnitude bank and bar trimming were largely offset by deposition of imported or scoured material during the discretized hydrograph run, thereby decreasing the overall magnitude of scour in those areas (Fig. 8B).

### 5.2.2 Path length distribution

We modelled annual-scale morphodynamics on the Feshie using five different path length distributions (Sect. 4.3.1). The similarity between the DEMs produced by the model (and hence, the DoDs shown in Fig. 10) using these distributions is striking, yet subtle differences may reflect the distinct nature of the spatial arrangement of erosion and deposition. Overall, the similarity between the modelled distributions may also be the result of the smoothing algorithms used in the model to ensure computationally-stable output surfaces, such as the along-flow averaging of shear stress and neighbourhood windows used for deposition (Sect. 2.4), both of which may act to reduce the variability introduced by the choice of a particular path length distribution.

In fluvial settings, erosional processes typically operate over small spatial scales (e.g. bank erosion, bar trimming, pool scour), and the magnitude of scour in these focused areas is typically higher than diffuse, broad-scale depositional processes such as overbank sheets or accretion of mid-channel or lateral bar material (Wheaton et al., 2013). As such, we suggest that the overall similarities, as well as the differences between our model's outputs using these contrasting path length distributions reflects the ability of deposition to counterbalance elevation changes due to scour of material, and the fact that the diffuse nature of the path length distributions used here make this counterbalancing difficult. For example, the compressed and stretched Gaussian distributions (Figs. 10A, 10B) are both marked by broad areas of erosion and deposition typically falling between ±1 m in elevation change. However, high-magnitude areas of erosion are more rare, yet still present, in the stretched Gaussian distribution, which may be due to the more longitudinally-extensive deposition of scoured material partially offsetting elevation changes due to erosion. The compressed and stretched Gaussian distributions stand in contrast to the exponential and shortened distributions (Figs. 10C - 10E), where elevation changes are largely confined between ±0.5 m, and overall are more fragmented across the model reach. This does not reflect a reduced magnitude of erosion, as Eq. (7) was used in all cases to predict scour depth; rather, the fragmented nature of elevation changes, along with the narrower range of those changes, is likely due to the propensity for deposition to offset erosional changes given the

more focused nature of the path length distributions in Figs. 10C - 10E. Nevertheless, in all distributions used, the volume of material deposited in a given cell following erosion upstream is always a fraction of that which was eroded. Using the distributions in Fig. 10 and the deposition smoothing window detailed in Sect. 2.4, the volume of sediment deposited falls between 0.4% and 8% of the volume which is eroded, and as such, erosion may outpace deposition in many cells.

### 5.2.3. Path-length modelling as compared to CA and CFD morphodynamics

While a full benchmarking comparison of our approach with other CA or CFD schemes is beyond the scope of this investigation, which solely seeks to demonstrate a novel hybrid approach to morphodynamic modelling, we can nevertheless draw general conclusions regarding the performance of this approach as contrasted with existing modelling efforts. We note that Williams et al. (2016a, 2016b) performed both CA hydraulic modelling using an approach similar to Murray and Paola (1994), and 2D CFD morphodynamic modelling (using Delft3D) on the same reach of the Rees River that was investigated using MoRPHED. At both low and high flows (7.3 $m^3$/s and 54.7 $m^3$/s, respectively; Williams et al., 2016a, Fig. 6), the CA hydraulic model over-predicted inundation extent and braiding intensity as compared to either field observations or CFD modelling, which were closely correspondent. The over-prediction by the CA model resulted from the sole reliance on bed gradient to route flow, as opposed to also incorporating the momentum components of the Navier-Stokes equations (Coulthard and Van de Wiel, 2013; Fonstad, 2013). Given that MoRPHED employed CFD-driven hydraulics (Section 2.1), and given that the boundary conditions employed in that modelling were largely identical to those used by Williams et al. (2016b), we hypothesize that the hydraulic modelling component of our approach on the Rees was likely in agreement with previous CFD modelling efforts conducted there.

With regard to morphodynamics, we can compare the general results of Williams et al. (2016b), who simulated the event-scale evolution of the same Rees River study reach used here. During a flood with a peak discharge of 227 $m^3$/s that occurred over a two-day period in April 2010 using Delft3D, Williams et al. (2016) computed 37,204 $m^3$ of scour, 27,692 $m^3$ of deposition, and a net sediment budget of -9,331 $m^3$, similarly thresholding sediment calculations at 0.1 m. Our own event-scale modelling of a four-day flood with a maximum discharge of 259 $m^3$/s in December of 2009 produced an estimated 28,297 $m^3$ of erosion, 28,102 $m^3$ of deposition, and a net sediment budget of -195 $m^3$. While these results are not directly comparable, given that they were generated from models of

two separate floods with varying (although similar) peak magnitudes, durations, and antecedent channel form, the first-order agreement between the magnitudes of geomorphic change for relatively similar event magnitudes provides evidence that our event-scale approach is able to produce similar magnitudes of geomorphic change as seen in CFD modelling that utilizes an Exner-based approach (i.e., Eq. 6) for computing morphodynamic evolution.

At the same time, divergence in the nature of geomorphic change between the two approaches is evident. While the majority of elevation changes computed by Williams et al. (2016b) were of low magnitude (e.g., ± 0.1 m) with respect to both erosion and deposition, MoRPHED predicted that the majority of sediment scour was the result of focused, high-magnitude erosion centered around -0.5 m (Fig. 7C). The depositional component of elevation change predicted by MoRPHED was marked by low-magnitude deposition comprising the majority of changes, in

line with the results of CFD modelling from Williams et al. (2016), thus suggesting that a path-length approach produces similar results with regard to deposition as compared to CFD morphodynamic modelling, whereas the computation of sediment scour on a once-per-event basis likely requires refinement. Finally, the parameterization of bank erosion proved challenging in both MoRPHED and in the Delft3D simulations of Williams et al. (2016b) on the Rees, although for differing reasons. In the case of CFD modelling, disparities between field observation

and model outputs primarily arose due to differences in the positioning and migration direction of individual channels (see Williams et al., 2016b Fig. 7), whereas in MoRPHED, model outputs generally predicted channel deepening in lieu of migration (Fig. 7E). Within both CFD and event-scale modelling, these results emphasize the need for further investigation into the mechanics of intra-flood bank erosion and lateral migration of braided rivers, as a function of both near-bank shear stress and slope of near-bank cells within the model domain. In particular,

Stecca et al. (2017) provide a framework for assessing the performance of noncohesive bank erosion algorithms and apply this within a cross-sectional framework. The application of this framework to identify the most behavioural algorithms for modelling bank erosion in two-dimensional simulations would be an appropriate next step to improve the model that is assessed here.

In our modelling approach, the parameterization of time as equal to a single event, or several points within one event, in the case of the hydrograph discretization case study (Sec. 4.2.1), stands in contrast to traditional CFD approaches, where time steps are represented by common units (i.e., seconds, minutes, hours, etc.). We recognize that this approach requires several simplifications, and perhaps chief among these is the notion that event-scale erosion is largely independent of event duration (see Section 2.2). For example, using the approach described here,

a flood of several days' duration would result in much less erosion than several short floods of the same magnitude, separated by periods of low flow, as the former would be modelled as a single event and the latter as multiple

discrete events. We made this simplification for two main reasons. First, laboratory and field research into particle travel distances during floods has developed robust relationships between channel morphology and event-scale deposition patterns (Kasprak et al., 2015; Pyrce and Ashmore, 2003a,b) and similarly computing event-scale erosion magnitudes, while inherently uncertain, across an area of interest provides a computationally-efficient way to utilize our growing knowledge of event-scale deposition dynamics. Second, in lieu of continuous intra-flood topographic surveys and/or intra-flood sediment monitoring at many points within a river, it is much more commonplace for geomorphologists to instead rely on information collected prior and subsequent to a competent flow. In effect, the modelling technique developed here operates on the same temporal scale as geomorphologists often work: before and after a geomorphically-effective flow. If such an approach is to prove useful for predicting fluvial morphodynamics, improvements in our ability to forecast event-scale erosion will be required, and this is an area deserving of further research. Similarly, this approach would benefit from an improved understanding of the timescales at which event-scale scour may be largely independent of event duration, versus those longer-duration events where scour is intrinsically linked to event duration. Certainly, for those floods of sufficient duration and/or magnitude to fundamentally alter bar spacing and thus invalidate the initial path length relationship used for modelling, adjustment of the modelling approach, specifically the user-defined path length distribution, is required at intra-flood timescales.

### 5.2.4 Imperfect models as exploratory tools

Models in the Earth sciences are necessarily imperfect (Oreskes et al., 1994), and the highly simplified nature of MoRPHED, combined with the highly dynamic and nonlinear nature of braided river morphodynamics (Ashmore, 1991; Bristow and Best, 1993) implies that our model will necessarily fail to achieve perfect replication of field-observed geomorphic dynamics. However, even imperfect models can provide meaningful insight into the processes behind morphologic evolution of fluvial systems (Paola et al., 2009; Paola and Voller, 2009). MoRPHED is designed to facilitate experimentation, particularly with regard to process inclusion or the particular aspects of process representation of bed and bank erosion, transport, deposition, and import dynamics (Fig. 1). For example, in Sect. 4.3.1, we explored the implications of altering the path-length distributions for bed and bank sediment transport/deposition, along with seeking to understand the advantages and drawbacks of discretizing hydrographs during model runs. The notion of morphodynamic modelling that employs sediment transport routines based on particle path length distributions is in its infancy, and we have built the model as an exploratory tool that can be used to investigate the utility of this approach toward predictive modelling of braided river evolution. Several

components of the model may be employed to investigate longstanding questions in our understanding of braided river dynamics, starting with the path-length approach itself. Recent field-based research on braided rivers has confirmed the coupled nature of sediment sources and sinks and the influence of sediment pathways on braiding maintenance (Williams et al., 2015). While it has long been hypothesized, and field and laboratory data have often confirmed, that mobilized particles in braided rivers are preferentially deposited in association with regularly-occurring channel bars (e.g. Pyrce and Ashmore, 2003a,b; Kasprak et al., 2015), the form of the path length distribution, and its relationship to geomorphic unit spacing, is deserving of further study across braided systems. As such, the choice of path length distribution and subsequent comparison of model results with field observations may provide insight into the applicability of path length distributions on a system-by-system basis (Hassan et al., 2013).

MoRPHED may also be used to investigate the utility of event-scale monitoring. Existing morphodynamic models typically employ a sediment continuity approach (e.g. Exner; Eq. (6), operating at very fine temporal scales, typically seconds to minutes. This approach produces results consistent with field observation (Bates et al., 2005), but comes at the expense of computational overhead, thereby restricting the timescales that can be modelled (Brasington et al., 2007). Because the timestep of the model is, by default, a single event, computational resources are conserved, allowing for extended simulations at annual and decadal timescales. However, it is unclear whether processes that occur over the course of a competent flow (e.g. avulsions, bank failures) can be adequately captured using an event-scale modelling approach. In lieu of a continuity-based approach for computing morphodynamic change, here we employed a simplified, event-scale estimation of bed scour depth as a function of (Eq. 7, Sect. 2.2). At present, it is unclear whether this approach is valid across a wide range of river styles and/or independent of event duration; one potential way forward in model calibration may be to scale Eq. 7 such that field-observed changes, particularly those seen in elevation change distributions at the event scale (Fig. 7), are matched as closely as possible, and then proceeding with modelling over longer timescales. Such a model calibration approach was not attempted here, but may improve event-scale modelling fidelity in future work. Similarly, the degree to which a hydrograph may need to be discretized and its constituent parts modelled (Sect. 4.2.1) to capture stage dependent processes such as the development of chute cutoffs or bar edge trimming (Wheaton et al., 2013) is deserving of further investigation.

Finally, we note that the sequencing of individual floods may have implications for the geomorphic evolution of modelled reaches using this approach. For example, a large flood event may result in widespread scour of the reach

and, under equilibrium sediment import parameters, may thus drive a large amount of bedload import and deposition at the upstream end of the reach prior to the next event being modelled. The choice of path length distribution used for imported sediment may further affect the degree to which large amounts of deposition could potentially occur at the upstream model boundary, and the implications of these choices on model fidelity, although not explicitly evaluated here, deserve further study. Finally, although periodic sediment-supplying floods have been surmised to maintain the braided planform in lieu of continuous sediment supply, and thus may allow braided rivers to persist under conditions of sediment deficit at annual scales (Wheaton et al., 2015), the influence of flood sequencing on reach-scale morphodynamics has received relatively little attention in the literature and is deserving of further work in the field as well as via numerical modelling.

Our exploratory research into decadal-scale modelling on the River Feshie (Sect. 4.4) indicates that both the accurate prediction of event-scale scour depth, and subsequent deposition location, present methodological hurdles in the development of valid morphodynamic models that operate at the timescale of competent flows. In our model, as in the field, erosion occurs in discrete, focused areas of high magnitude (e.g. pool scour, bar trimming, bank retreat; Ashmore, 1991; Bristow and Best, 2003; Wheaton et al., 2013). Deposition occurs thinly over more broad spatial areas (e.g. bedload sheets, overbank sheets, bar formation), a result of the path length distributions employed here and the smoothing required to avoid the generation of rough topography that would lead to hydraulic instability. One result of the differences in the nature of erosion and deposition is that deposition may never 'catch up' to erosion if a simple path-length distribution is employed, thus resulting in over-scouring of channels (Fig. 11). In practice, this emergent behaviour can be seen in the large-scale downcutting seen in the central anabranch on the Feshie over the decadal run from 2003-2013 (Fig. 11, Sect. 4.4). It is possible that this artefact may simply be the result of the differing algorithmic nature of erosional processes versus depositional ones as represented in the model (e.g., discrete versus dispersed). On the other hand, it is well known that erosion and deposition carry divergent spatial signatures in the field (Wheaton et al., 2013), and it is also possible that the over-scouring of existing channels may arise from a lack of bed sediment armouring in the model, improper representation of bank erosion magnitudes, as discussed below, over-dispersion of deposited sediment as described by path length distributions, or a combination thereof. As opposed to scour, the morphodynamic signature of deposition generally mirrors that seen in the field (see ECDs in Figs. 7, 9, 11), with a large contribution of total change coming from areas of shallow deposition. However, in future event-scale morphodynamic models, it may be necessary to augment path length distributions so as to preferentially deposit material in certain geomorphic units (e.g.

confluence pools, deep channels adjacent to banks) in order to develop lateral flow, bank material removal, and channel migration/avulsion (Ashmore, 1991).

Another process that is difficult to capture algorithmically is the lateral retreat of banks. Highly erodible banks are a hallmark of braided rivers and lead to the development of central bars and multiple anabranches (Ashmore, 1991). However, the Cartesian grid employed in the model, along with the event-scale timestep of the model, makes the generation of smooth bank features difficult. While approaches are available that compute bank stability based on a factor-of-safety approach that balances downslope gravitational forces with the ability of bank material to provide cohesive resistance to failure (Darby and Thorne, 1996; Simon et al., 2000; Rinaldi and Darby, 2007), parameterization of these models, especially at the reach scale, is quite difficult. The simplified approach employed in the model averages the slope of bank cells and the near-bank shear stress of the flow to predict bank retreat distances (Sect. 2.3). The threshold slope and shear stress, and their effect on lateral retreat distance, have been empirically adjusted to emulate field-observed bank dynamics. We have observed that the simple treatment of lateral erosion in the model produces bank erosion and bar edge trimming. However, whether this simplified approach will provide computational stability over longer-term (e.g. centennial) simulations, when bank retreat is only computed once per flood, is unknown. Additionally, further investigation is needed to determine whether the inclusion of bank toe deposition, as opposed to immediate downstream transport according to a user-specified path length distribution, is necessary in the model, along with whether bank material should be transported and deposited according to the same path-length distribution as was used for bed material.

## 6 Conclusions

The morphodynamic model developed here simulates braided river evolution at a variety of timescales via a path-length based approach, and here we applied the model to two braided river reaches at the event, annual, and decadal scale. The premise of MoRPHED is that sediment transport can be approximated using a steady flow set to the event peak discharge and path length distributions (Pyrce and Ashmore, 2003a,b; Kasprak et al., 2015), which, when coupled with two-dimensional hydraulic simulations at peak flood discharge, results in decreased computational overhead, thus enabling longer simulations at improved spatial resolution. We observed overall correspondence between model outputs and field observations when comparing planform changes, geomorphic change described by elevation change distributions, and morphometric indices such as sinuosity, braiding index, and channel node counts (Sect. 2.7.1). At the same time, divergence between field and model datasets was evident,

suggesting that a purely path-length-based approach may oversimplify the highly dynamic nature of braided systems (Bristow and Best, 1993). Specifically, at event scales on the Rees River, New Zealand, we observed under-prediction of areal and volumetric changes (total areal extent of change in the field was 287,184 $m^2$ compared to 142,547 $m^2$ in the model, total volumetric change was 83,149 $m^3$ in the field and 36,851 $m^3$ in the model). At

annual timescales on the River Feshie, UK, the model over-predicted the extent and volume of geomorphic change (areal extent of change was 17, 206 $m^2$ in the field versus 37,707 $m^2$ in the model, volumetric changes were 4,349 $m^3$ in the field and 10,149 $m^3$ in the model). When conducting decadal-scale modelling on the Feshie over the period from 2003-2013, we observed good correspondence between areal and volumetric predictions of deposition (areal extent of deposition was 36,045 $m^2$ in the field compared to 26,861 $m^2$ in the model, volumetric deposition

was 14,677 $m^3$ in the field and 13,748 $m^3$ in the model) and the areal extent of erosion (23,836 $m^2$ in the field compared to 23,542 $m^2$ in the model), but the model overpredicted the volume of erosion throughout the reach (12,680 $m^3$ in the field compared to 23,744 $m^3$ in the model).

While we did observe reproduction of all field-observed braiding mechanisms (Wheaton et al., 2013), the relative

contribution of these mechanisms often varied from values seen in the field. In contrast to all other braiding mechanisms, neither bank erosion nor bar edge trimming emerged simply as a result of bed scour and deposition, and instead needed to be explicitly parameterized in the model. In addition, chute cutoff only occurred at areas where pre-existing chutes or channel heads were observed, and did not appear to emerge across previously flat bar tops. While this model represents a first step in event-scale, path-length-based morphodynamic modelling, further

testing is needed to evaluate the feasibility of the approach for longer-term modelling runs and/or for determination of whether process representation that accounts for inter-flood geomorphic change, such as avulsions or bank mass failure (Leddy et al., 1993; Ashworth et al., 2004) will require explicit parameterization. We have designed the model to be a modular framework for exploring the effect of various process representations, and as a learning tool designed to reveal the relative importance of geomorphic transport processes in braided river dynamics at multiple

timescales.

**Data Availability:** All model code used in this manuscript can be found at https://github.com/morphed/MoRPHED.

**Competing Interests:** The authors declare that they have no competing interests.

**Acknowledgements:** This research was supported by a grant from the National Science Foundation (No. 1147942). We thank Philip Bailey (North Arrow Research) for extensive assistance with model code and algorithm development. Field work on the Feshie was completed with the generous permission of Thomas MacDowell and the Glenfeshie Estate, with the assistance of Niall Lehane (Queen Mary University of London), Mark Smith (University of Leeds), Julian Leyland (Southampton University), and Damiá Vericat (Forest Technology Institute of Catalunya). Our modelling efforts benefited from substantial discussion with Sara Bangen, Nate Hough-Snee, Wally MacFarlane, Eric Wall, and Peter Wilcock (Utah State University), along with Rebecca Hodge (Durham University) and David Sear (Southampton University). The comments of three anonymous reviewers greatly improved the quality of this manuscript, and we thank them for their contribution.

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

**FIGURE CAPTIONS**

**Figure 1.** MoRPHED model operation flowchart

**Figure 2.** Lateral migration algorithm schematic

**Figure 3.** Particle transport and deposition routine. (A) shows computation of sediment to deposit as a function of eroded sediment via a particle path length distribution, (B) shows neighbourhood window approach for averaging deposited sediment over adjacent cells.

**Figure 4.** Path length distributions used in MoRPHED modelling on (A) the River Feshie and (B) the Rees River. Peak of the distributions corresponds to average along-flow spacing between confluence and diffluence pairs.

**Figure 5.** Morphodynamic modelling sites. Overview maps of Rees River (left) and River Feshie (right). Hillshaded DEMs (2 m resolution for Rees and 1 m resolution for Feshie) are shown atop aerial photograph base. Information on data availability for both sites shown in lower panels.

**Figure 6.** River Feshie hydraulic modelling. Surveyed water surface extent in each of five years (2003-2007) was compared to modelled inundation extent in the same years using discharge levels approximated using data from gauge at Feshiebridge. Areas observed (but not predicted) to be inundated shown in blue, areas predicted (but not observed) to be inundated shown in green. Areas which were correctly predicted as being inundated shown in red. Base is hillshaded 1 m DEM.

**Figure 7.** Rees River event morphodynamic modelling results. Continuous hydrograph and modelled discharge shown in (A). Elevation change distributions derived from field and model data shown in (B) and (C) respectively. Braiding index and total sinuosity for field and model results shown below (A); numbers in parentheses indicate the difference between initial and final observations obtained from field surveys and modelling. DoD derived from field data shown in (D), with model-derived DoD and delineated braiding mechanisms shown in (E) and (F). Volumetric contributions of each braiding mechanism in field and modelled data shown in (G); colours in (F) and (G) are identical for each braiding mechanism.

**Figure 8.** Rees River discretized hydrograph case study. DoD from MoRPHED modelling shown in (A), with ECD in (B),event hydrograph with model points shown in (C), and volumetric contribution of braiding mechanisms in field and model shown in (D). Refer to field-based results shown in ECD (Panel B) and DoD (Panel D) in Figure 7 for comparison.

**Figure 9.** River Feshie annual morphodynamic modelling results. Refer to Fig. 7 caption for details.

**Figure 10.** River Feshie variable path length case study. Path length distributions used are: compressed Gaussian (A), stretched Gaussian (B), and exponential (C), along with shortened Gaussian (D) and shortened exponential

10    (E). DoDs for each path length distribution shown in middle row, with elevation change distributions shown in bottom row.

**Figure 11.** River Feshie decadal morphodynamic modelling results. Refer to Fig. 7 caption for details. Note variable axes limits in (B) and (C).

**TABLE 1.** Delft3D hydraulic modelling parameters.

| Site | Sim. Time (hrs) | Time Step (s) | Cell Size (m) | Nodes | $D_{84}$ (m) | C-W Roughness ($k_s$) | Eddy Viscosity ($v$, m$^2$s$^{-1}$) |
|------|-----------------|---------------|---------------|-------|--------------|------------------------|--------------------------------------|
| Rees | 1 | 0.025 | 2 | 352,231 | 0.035 | 0.10 | 0.1 |
| Feshie | 1 | 0.025 | 1 | 201,116 | 0.1 | 0.29 | 0.1 |

**TABLE 2.** Rees River event modelling: geomorphic change detection results; results of discretized modelling shown in *italics*.

| | *Field Raw* | *Field Thresholded (0.1 m)* | *MoRPHED Raw* | *MoRPHED Thresholded (0.1 m)* |
|---|---|---|---|---|
| **Areal** | | | | |
| Total Area of Erosion (m²) | 435,604 | 146,472 | 390,555 *418,856* | 57,387 *63,496* |
| Total Area of Deposition (m²) | 461,084 | 140,712 | 500,233 *389,072* | 85,160 *94,008* |
| **Volumetric** | | | | |
| Total Volume of Erosion (m³) | 57,324 | 47,598 | 28,297 *28,679* | 20,663 *20,291* |
| Total Volume of Deposition (m³) | 47,022 | 35,551 | 28,102 *28,130* | 16,188 *19,480* |
| **Vertical Averages** | | | | |
| Average Depth of Erosion (m) | 0.13 | 0.32 | 0.07 *0.07* | 0.36 *0.32* |
| Average Depth of Deposition (m) | 0.10 | 0.25 | 0.06 *0.07* | 0.19 *0.21* |
| Average Total Thickness of Difference (m) | 0.12 | 0.09 | 0.06 *0.07* | 0.04 *0.05* |
| Average Net Thickness of Difference (m) | -0.01 | -0.01 | 0.00 *0.00* | -0.01 *0.00* |

**TABLE 3.** River Feshie annual modelling, geomorphic change detection results.

| | *Field Raw* | *Field Thresholded (0.1 m)* | *MoRPHED Raw* | *MoRPHED Thresholded (0.1 m)* |
|---|---|---|---|---|
| **Areal** | | | | |
| Total Area of Erosion ($m^2$) | 66,074 | 9,970 | 50,021 | 17,444 |
| Total Area of Deposition ($m^2$) | 49,998 | 7,236 | 58,366 | 20,263 |
| **Volumetric** | | | | |
| Total Volume of Erosion ($m^3$) | 4,433 | 2,806 | 6,149 | 5,239 |
| Total Volume of Deposition ($m^3$) | 2,704 | 1,543 | 5,986 | 4,910 |
| **Vertical Averages** | | | | |
| Average Depth of Erosion (m) | 0.07 | 0.28 | 0.12 | 0.30 |
| Average Depth of Deposition (m) | 0.05 | 0.21 | 0.10 | 0.24 |
| Average Total Thickness of Difference (m) | 0.06 | 0.04 | 0.11 | 0.09 |
| Average Net Thickness of Difference (m) | -0.01 | -0.01 | 0.00 | 0.00 |

**TABLE 4.** River Feshie decadal modelling, geomorphic change detection results.

| | *Field Raw* | *Field Thresholded (0.2 m)* | *MoRPHED Raw* | *MoRPHED Thresholded (0.2 m)* |
|---|---|---|---|---|
| **Areal** | | | | |
| Total Area of Erosion (m$^2$) | 38,332 | 23,836 | 48,092 | 23,542 |
| Total Area of Deposition (m$^2$) | 70,329 | 36,045 | 57,087 | 26,861 |
| **Volumetric** | | | | |
| Total Volume of Erosion (m$^3$) | 13,944 | 12,680 | 25,134 | 23,744 |
| Total Volume of Deposition (m$^3$) | 18,264 | 14,677 | 15,712 | 13,748 |
| **Vertical Averages** | | | | |
| Average Depth of Erosion (m) | 0.36 | 0.53 | 0.52 | 1.01 |
| Average Depth of Deposition (m) | 0.26 | 0.41 | 0.28 | 0.51 |
| Average Total Thickness of Difference (m) | 0.30 | 0.25 | 0.39 | 0.36 |
| Average Net Thickness of Difference (m) | 0.04 | 0.02 | -0.09 | -0.10 |

**Figure 1.**

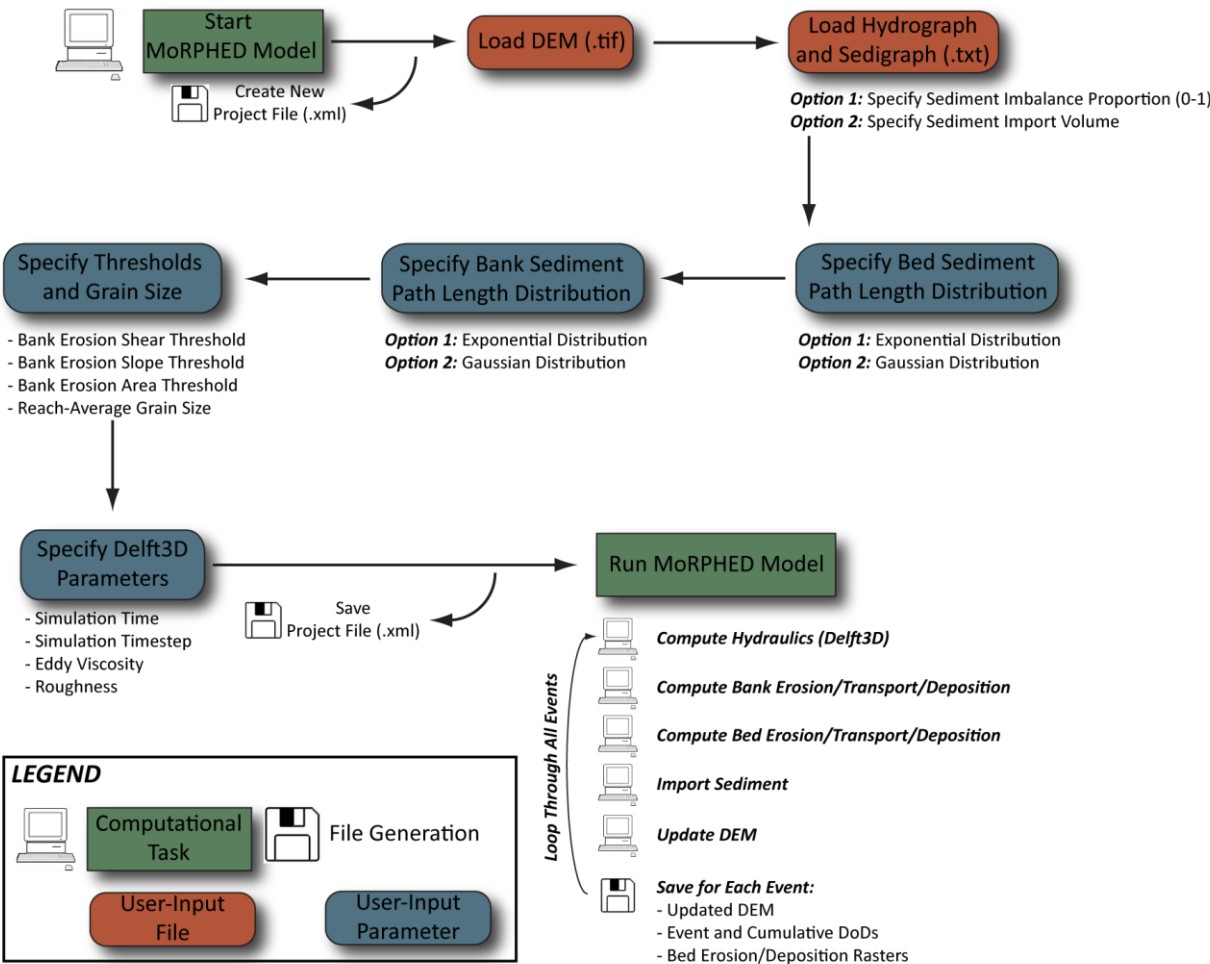

**Figure 2.**

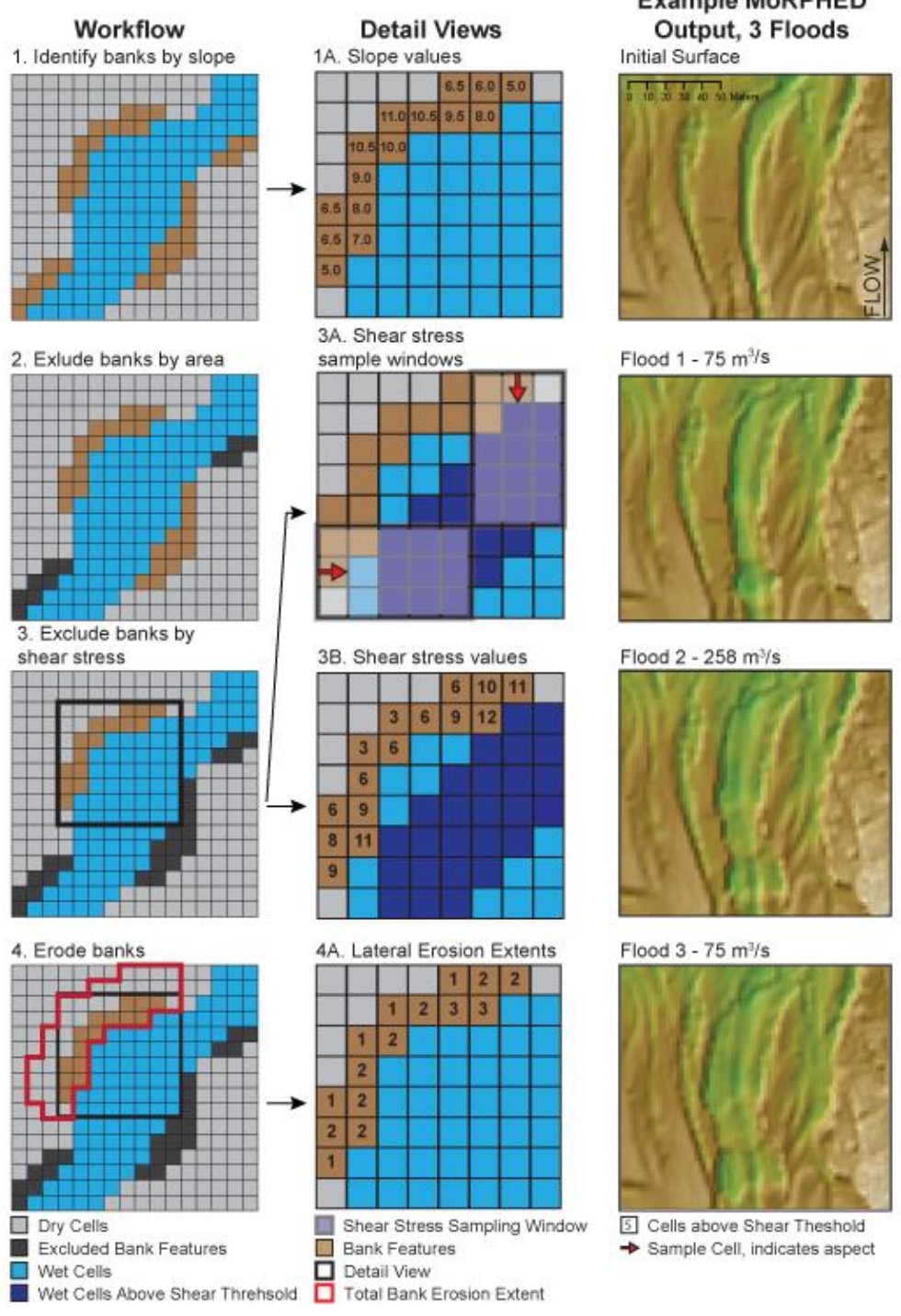

**Figure 3.**

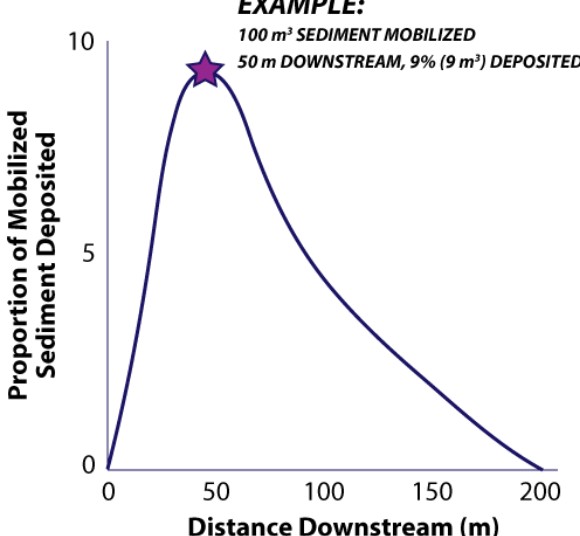

**A.**

EXAMPLE:
100 m³ SEDIMENT MOBILIZED
50 m DOWNSTREAM, 9% (9 m³) DEPOSITED

*Proportion of Mobilized Sediment Deposited*

*Distance Downstream (m)*

**B.**

TOTAL SEDIMENT TO DEPOSIT: 9 m³

| | | | | |
|---|---|---|---|---|
| 0.1875 m³ | 0.1875 m³ | 0.1875 m³ | 0.1875 m³ | 0.1875 m³ |
| 0.1875 m³ | 0.375 m³ | 0.375 m³ | 0.375 m³ | 0.1875 m³ |
| 0.1875 m³ | 0.375 m³ | 3.00 m³ | 0.375 m³ | 0.1875 m³ |
| 0.1875 m³ | 0.375 m³ | 0.375 m³ | 0.375 m³ | 0.1875 m³ |
| 0.1875 m³ | 0.1875 m³ | 0.1875 m³ | 0.1875 m³ | 0.1875 m³ |

**Figure 4.**

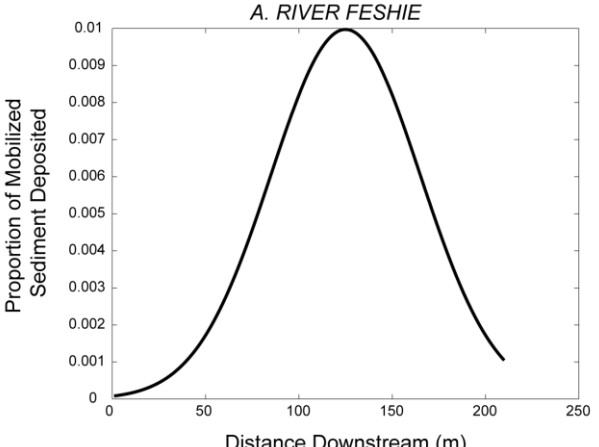
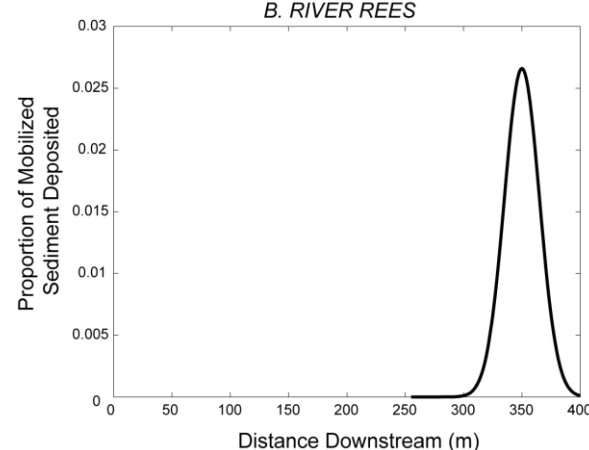

**Figure 5.**

River Rees, New Zealand

River Feshie, Scotland

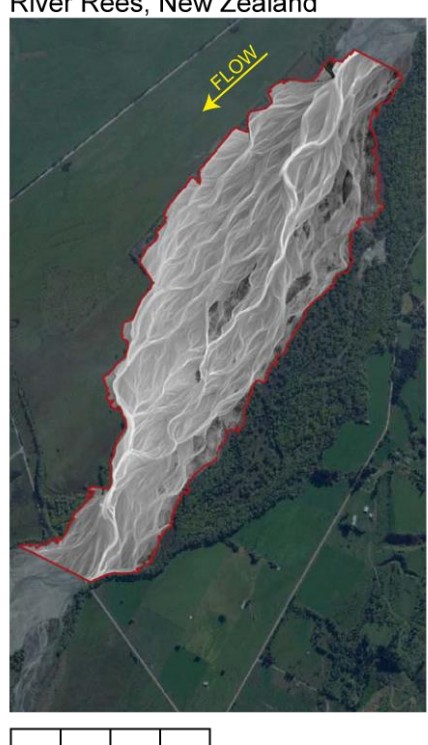

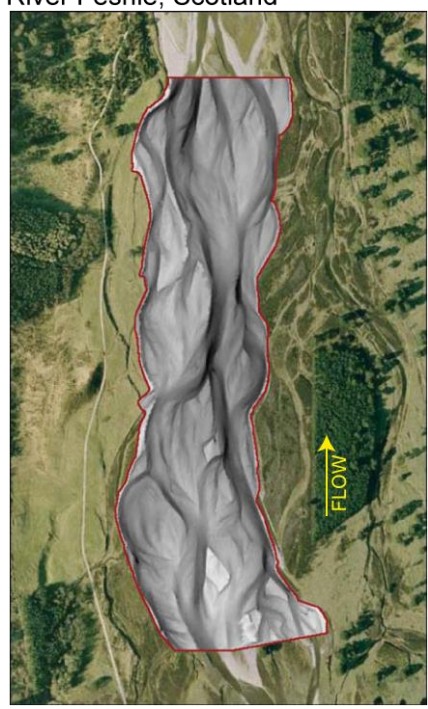

0    200   400   600   800 Meters

0    50    100   150   200 Meters

**Available Data: Rees**

Event-Annual Scale:
 - Resurveys Following 10 Events
   over a 2-Year Timespan

- 15-Minute Hydrograph Record

- Existing Delft3D Hydraulic
  Modeling (Williams et al., 2013)

**Available Data: Feshie**

Annual-Centennial Scale:
 - Resurveys 2003-2007 & 2013

- 15-Minute Hydrograph Record

- Aerial Photographs since 1946

- Planform Maps Since 1869

**Figure 6.**

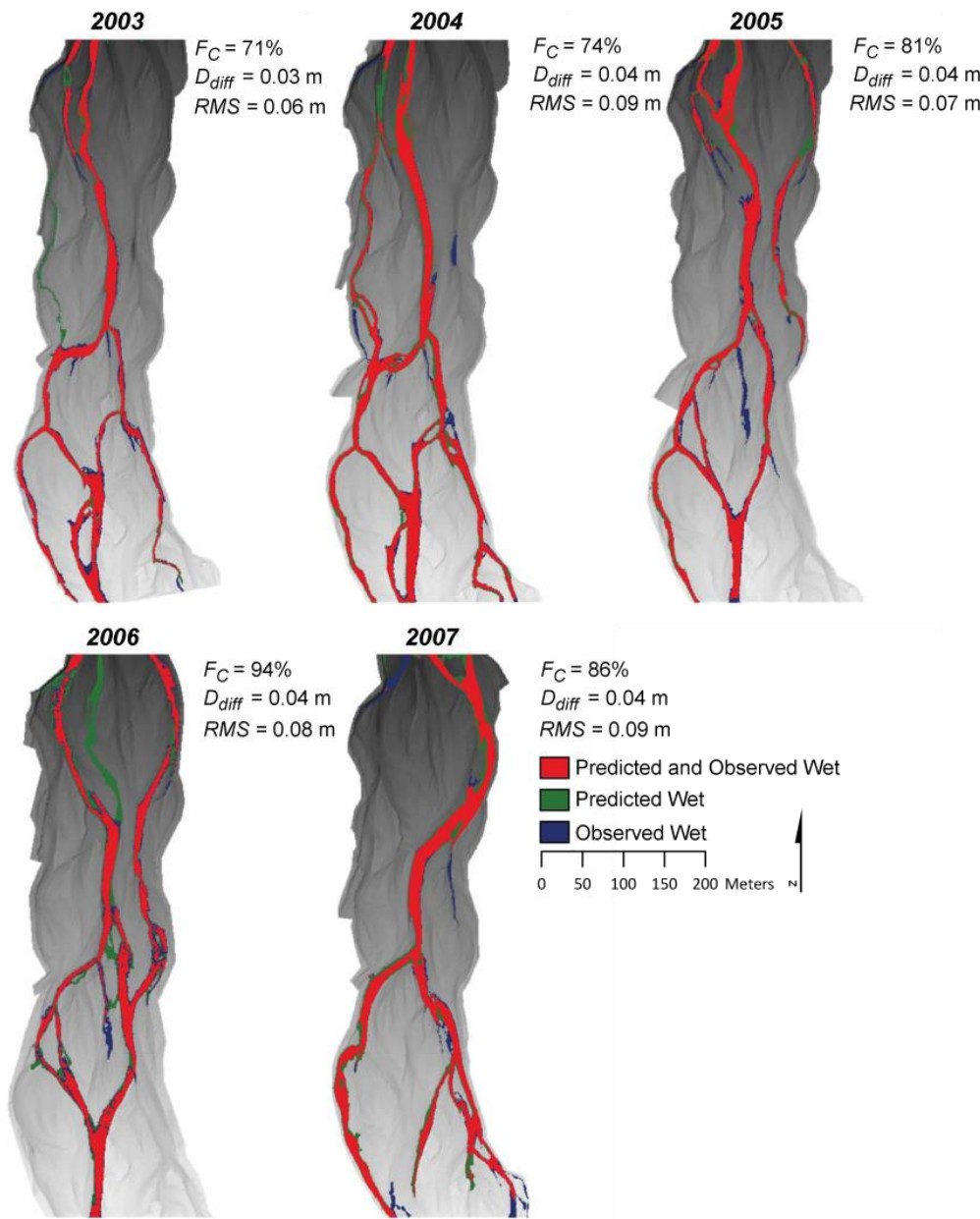

**Figure 7.**

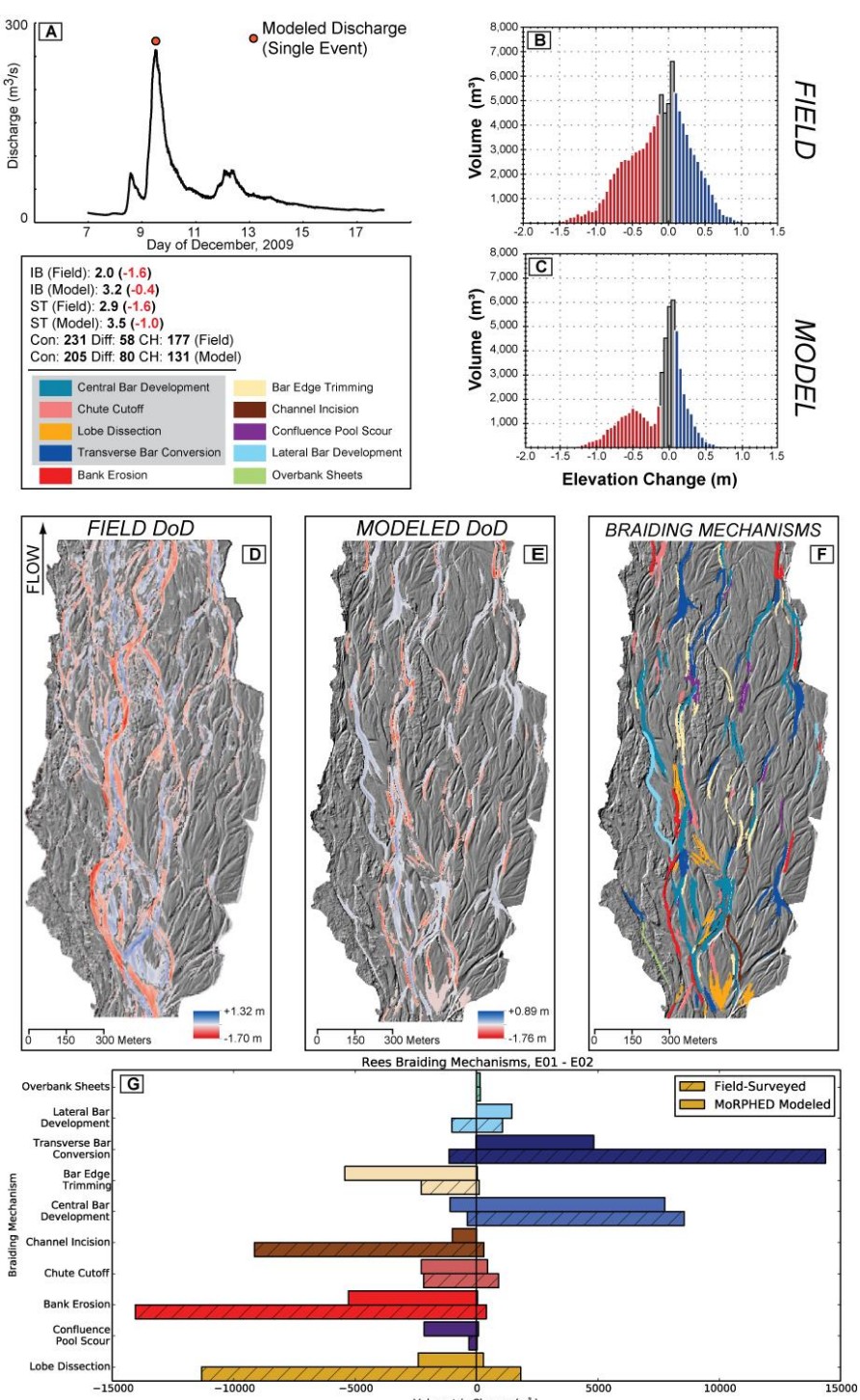

**Figure 8.**

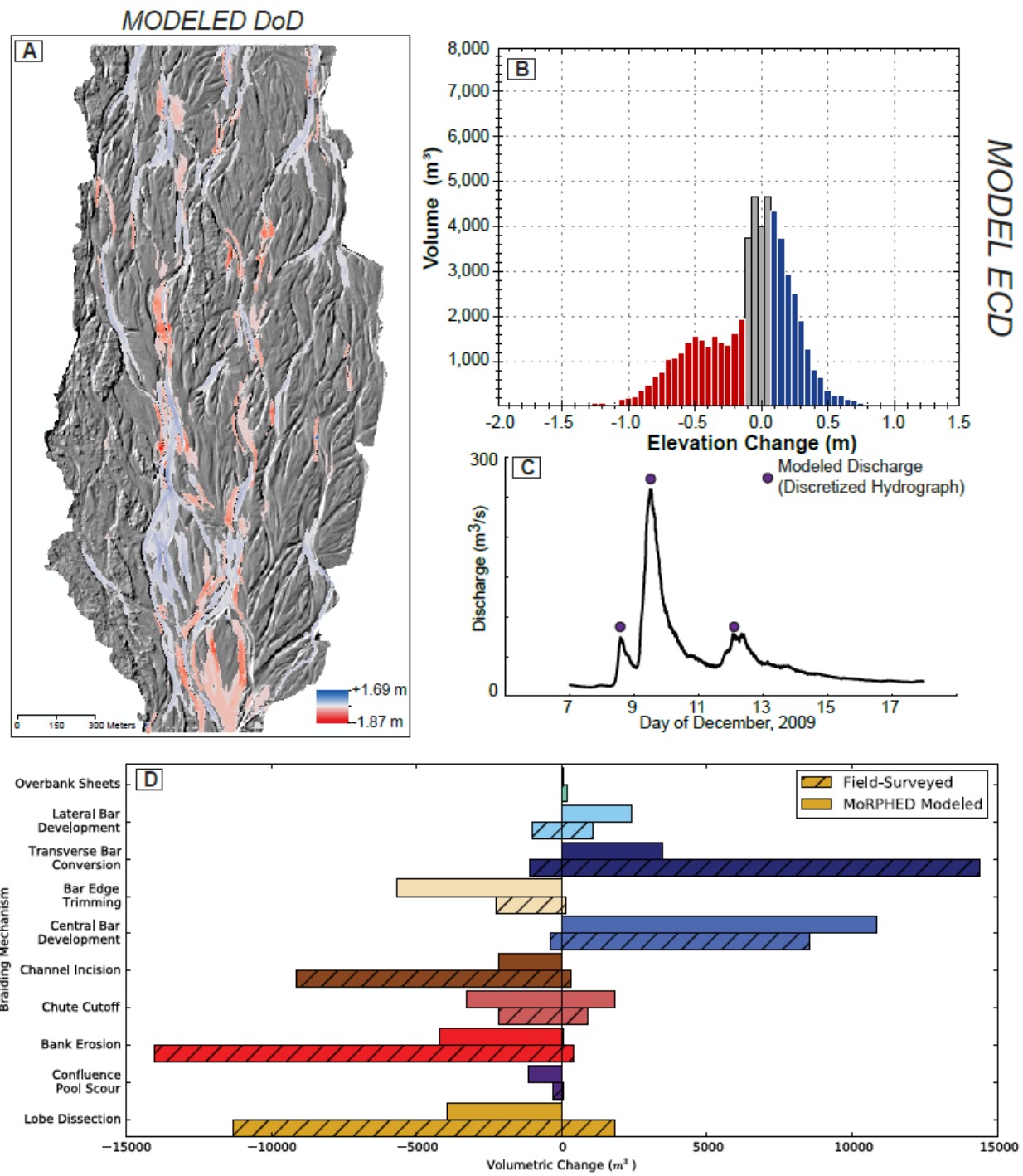

**Figure 9.**

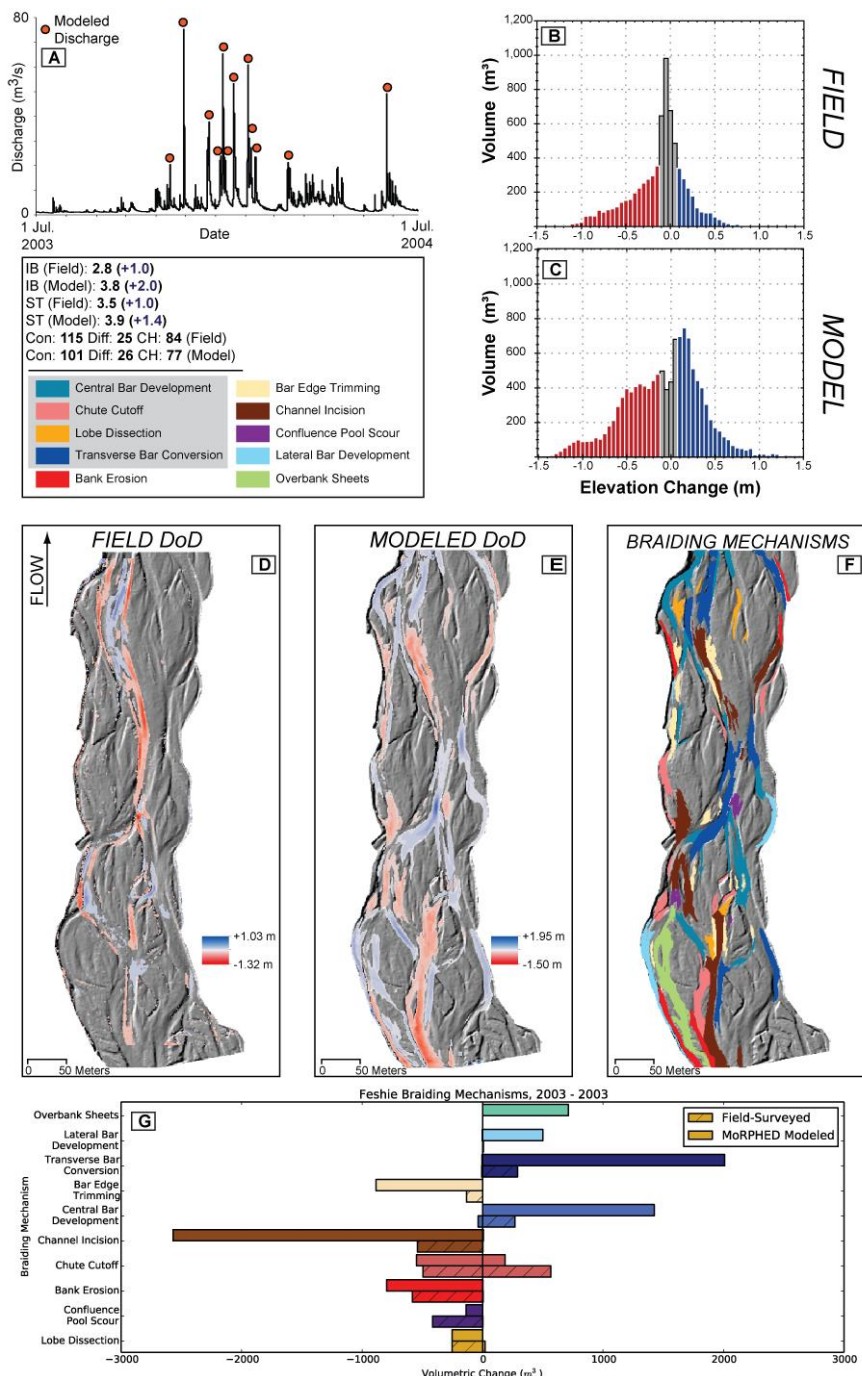

**Figure 10.**

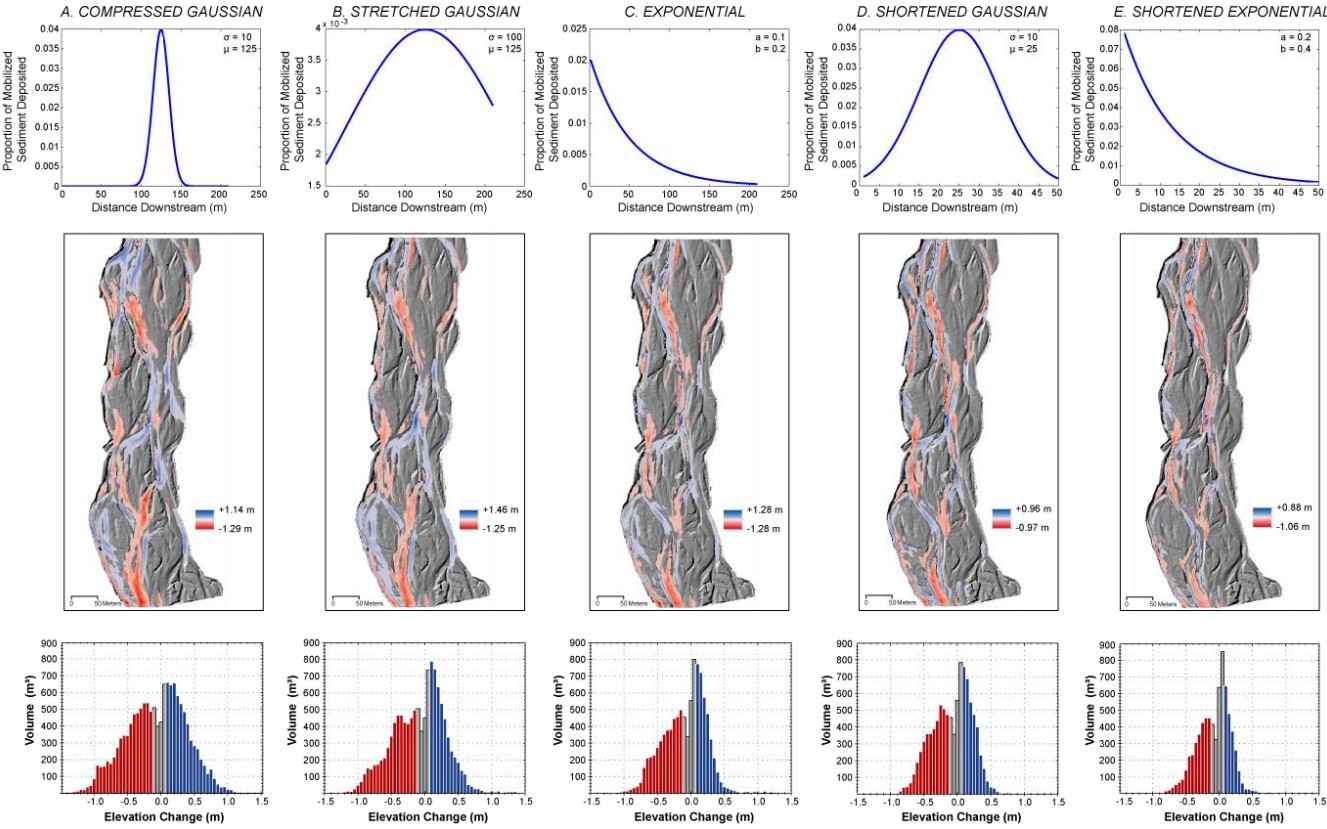

**Figure 11.**

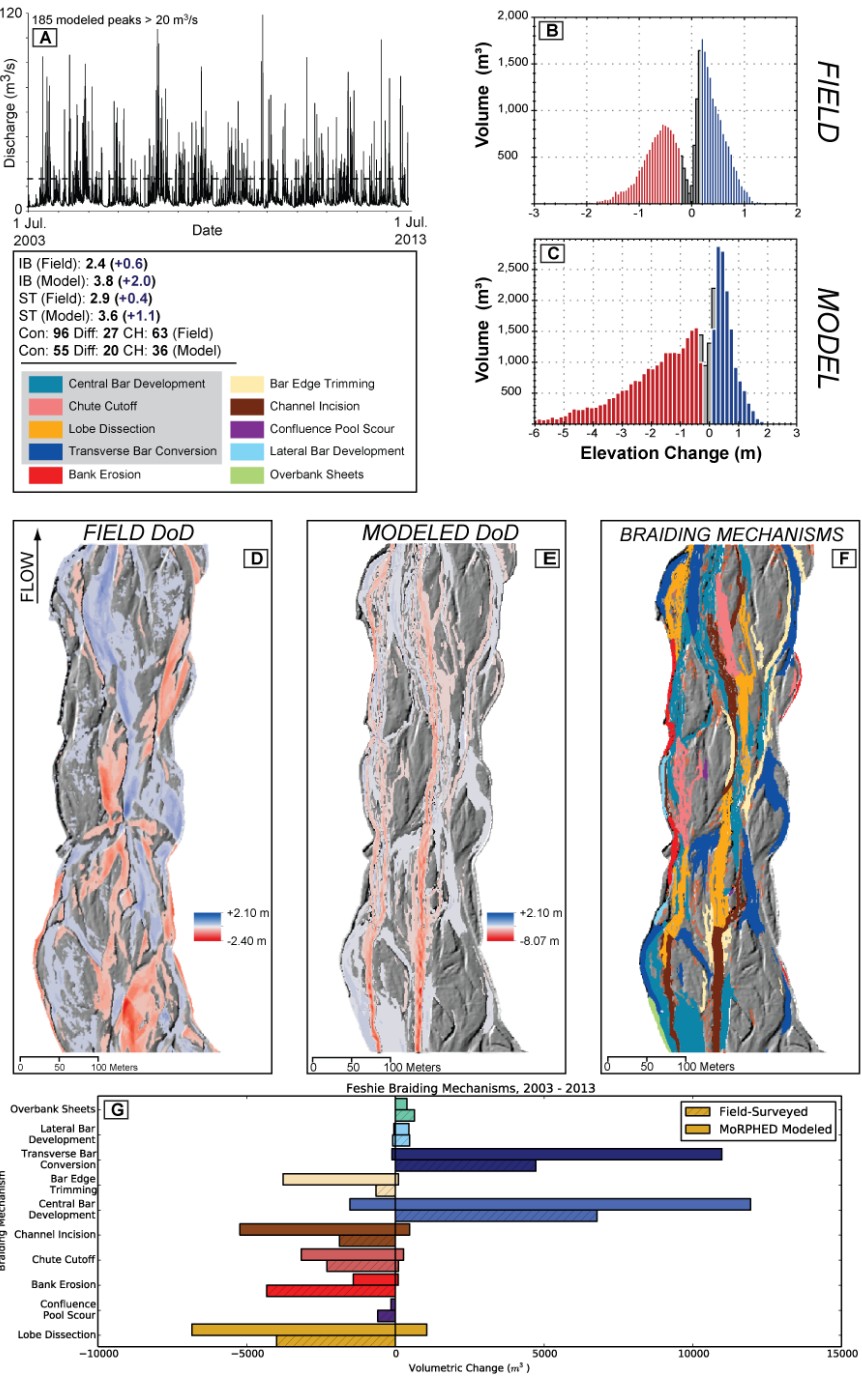