# Peer review of "Modelling braided river morphodynamics using a particle travel length framework"

_Earth Surface Dynamics, 2018_

## Referee Comment (RC1) · Anonymous Referee #1 · 6 May 2018

The manuscript presents an effort of modelling braided river morphodynamics by combining a 2D hydrodynamic model and a path-length based algorithm. The topic is relevant to an important issue on the earth surface dynamics, and it is should be interesting to readers of this journal. The aim of present hybrid approach is to develop a new framework of model to predict the braided river processes with limited computation time. Model predictions have been compared with two natural braided river for multi-scalar verification. The object of this study is clear. However, basic method is not described sufficiently, and then it is difficult to estimate the value of the new model. In addition the prediction is not close enough to the measurement even in the statistical meaning, and the discrepancy is not explained enough. Therefore, in my opinion, the current version of this manuscript should be improved before being accepted by the

journal for a publication. The specific comments are as following: 1. Page 5, Lines 21-25, Section 2.1: It is not clear if time derivatives are included in the model? In the model description, "Delft3D solves the shallow-water form of the Navier-Stokes equations, which related changes in momentum (left-hand terms) in time and space . . .", while the terms of time derivative are not included in Equations 1-2. 2. Page 6, Lines 1 & 8-9: "Where x and y, respectively, denote the streamwise and cross-stream directions of velocity (u, v), . . ." Does the "streamwise" mean the river direction? Because the "For all modelling, we employed fixed Cartesian orthogonal grids", and the velocity changes over time and space. It is unclear whether the x and y is in a Cartesian coordinate system or other curvilinear coordinate system? 3. Page 8, Lines 15- 20: This part is the key solution of the proposed model framework, however, it cannot be understand clearly how to calculated the morphological change rate by using Equation 7 and with the concept of "integrated sediment transport pathways". More description and comments are needed at least here. Without a confirmed understanding, the value of the new model cannot be estimated sufficiently and the results would not be analyzed correctly. 4. Page 9, Lines 25-30: It is not clear, when a bank erosion would occurred. Is there any relationship between the "7%" and angle of repose? What is the mean of "we set this threshold area to 30 cells, again adjusting this value to mirror the size of field-observed bank erosion patch"? This part is a key procedure for the river migration, however the meanings are too ambiguous. 5. Page 10, Section 2.4: The general meaning is not clear for this section. Firstly, "velocity vectors need not pass through the centre of each cell." Why? And what is the result for the "Bed/Bank Sediment Transport and Deposition"? Secondly, how to combine the 5x5 deposition window of cells and the path length distribution in the model? Comments and figures should be used to illustrate the complex an core relationships. 6. Page 12, Line 5: How to determine the Z_ACT? 7. Page 13, Lines 25-30: DoD means "Difference of DEM" or "DEM of Difference" or others? 8. Page 18, Line 30: There is only a single sub-section 4.2.1 in this section. 9. With regard to results and discussion, it is difficult to assess the agreement and disagreement of predictions without fully understanding of the model.

**ESurfD**

Interactive
comment

---

## Referee Comment (RC2) · Anonymous Referee #2 · 13 May 2018

Review of paper "Modelling braided river morphodynamics using a particle travel length framework", submitted to ESurf by Kasprak et al.

Overview

This is a well written exposition of a novel hybrid morphodynamic modelling approach that seeks to overcome the CPU-time limitations of conventional physics-based numerical models, thereby enabling the simulation of braided rivers over extended time and spatial scales. The CPU-time savings accrue from setting the morphological time step at the event scale. The event hydrograph is represented by a single discharge for which a robust steady-state hydrodynamic solution is derived. This is used to estimate entrainment from bank erosion and bed scour, with the entrained sediment then advected downstream using an assumed distribution for particle travel length and also

dispersed laterally. The approach is demonstrated using the case examples of the braided Rees and Feshie Rivers, with comparisons made between surveyed and modelled DEMs-of-difference in regard to volume-weighted elevation change distributions and classifications of braiding process. Nested "experiments" are also run that compare the model results when an event is represented by several quasi-steady flows rather than one, and when different types of path-length distributions are assumed. Extended discussion examines the limitations observed with the model results and around model assumptions, with good pointers on where to focus improvements.

Specific comments

1. On page 7 lines 1-6 nothing is said about the duration of the steady upstream discharge (which is set at the peak discharge). Surely, this is critical for scaling the volume and at least the erosion depth axes on the ECDs, and for matching these to the surveyed ECDs. How was this set?

2. Pursuing this further, applying a scaling-factor to the bedload transport function would induce similar effects to varying the event "duration", so what about using this concept as a formal calibration approach – adjusting the duration or BT function to align model ECDs to the few observed field results, then running the so-calibrated model for longer times scales? It seems that so far, the paper only considers using the Rees and Feshie field data for validation purposes, not calibration.

3. On page 7, around lines 20-25, it appears that only the median grain size is used for calculating the threshold stress, which is then used to calculate the bedload transport with equation 8 (for use in equation 7). While the method is claimed to be multi-fractional, there is nothing said about hiding factors, partial transport, different path-length distributions by grainsize, etc.). Equal-mobility is implicitly assumed, so effectively, entrainment and deposition are undertaken only considering the median size, so (apart from remixing of the active and sub-surface layers at the end of the event, as explained on page 30) the method doesn't actually do multi-fraction processes. This

means that it will not produce armouring, which can be an important control on scour depth. So, claiming the approach is multi-fraction is not true – it only pays lip-service to being multi-fraction.

4. On page 8, line 2 -3, it's not clear if 10 cells are being taken upstream and another 10 cells downstream or if 10 cells are being taken centred on the current cell. If the former, then in the case of the Rees mode,l 40 m averaging appears too excessive. Please clarify and justify if need be.

5. On page 8, lines 15-18, the logic around the alternative to the Exner equation is never completed. What appears to be done (but is not properly explained) is that "clear-water" scour is first calculated everywhere it can potentially occur, with a Ds assigned to each cell, then a deposited sediment thickness is assigned to cells downstream of erosion sites according to the path-length distribution and smearing algorithms. Thus, some cells will only experience scour, some only deposition, and some that scour will also experience deposition from sediment sources from scour sites upstream. Net elevation change at-a-cell is therefore calculated as the difference between scour and deposition. This needs to be explained, and if I have the story wrong, the right story needs explaining please.

6. On pages 10-11, a diagram (similar in concept to Fig 2) is needed to help explain the distribution of sediment deposition.

7. Page 14, lines 21-41, the methodology used for classifying braiding mechanisms is not stated. Was this purely subjective and done manually, or were algorithms employed that obeyed a set of rules? What were the guiding rules? Was the same approach applied to the field data and the model output?

8. Page 16, lines 6-9. Isn't it more a case that the Rees, by virtue of its frequent runoff events and significantly finer bed material (from its schist catchment), is a lot more labile which inhibits woody vegetation (and associated fine sediment trapping) from establishing?

9. Section 4.1.2 addresses only validation results, since no calibration was actually done. So, in line 11 on page 17 replace "calibration" with "validation". Also, on page 17 around lines 19-22 it would be useful to mention the Ddiff and Fc numbers given in the Figure. Also, I recommend also providing the RMSE of depth differences. The mean Ddiff statistic reported only informs on any overall bias in depth and tells nothing about how badly the depth may have been predicted locally.

10. Section 4.2, page 17-18, several things. First, some more information is needed here to clarify what was done in the modelling – particularly confirm what discharge was used and how long it was run for. Second, no mention is made at all in the text about thresholding but it is the thresholded results of volume change that are being discussed, not the gross ones. I would expect to see some comparison and discussion, particularly since the modelling doesn't have thresholding issues. Third, in lines 22-23 the statement that "average magnitudes of erosion and deposition agree well between field and model results" is a bit rose-tinted, because the differences appear to be in the range of 33-50%. Fourth, it would be very informative to compare the MoPHED-modelled results for the Rees event with the results of the full Delft3D model run of the same event (already done by Williams et al). Fifth (and again), what were the durations of the steady-flow model runs, how were they decided, and how do they impact on the results (I am assuming that when "model run time" is mentioned that this is processing time, not event duration). It would be useful to show the discretised quasi-steady hydrograph time-spans for the single event discretisation and for the 3-event discretisation on top of the hydrograph in Fig 7A.

11. Page 19, lines 20-29. Isn't all this just saying that you picked events when the flows were competent as indicated by Ashworth and Ferguson's observed threshold? The "low bankfull" flow story seems like an unnecessary complication.

12. Page 22, line 10. I disagree with this stated general agreement in form of the ECD. The model-predicted ECD is clearly more erosion skewed than what was observed. Indeed, what the model appears to have done with these prolonged runs is to cut

straight, deepening "ditches". I've experienced this before with long Delft3D runs (e.g. Singh et al), and I suspect it stems from inadequate handling of bank erosion and probably also lack of armouring. The model has also produced net degradation that is 2+ times the magnitude of the net aggradation that was observed in the field. So, there is an un-natural emergent behaviour occurring here in the model, which needs to be discussed – and hopefully dealt to with some clever workaround. It would be useful to provide the changes in the sinuosity over the decade of model runs to quantify this trait.

13. Page 23, line 25-26. Yes, this is an important point – the latency between flow, sediment transport and morphologic response is not captured by the quasi-steady approximation, so transient features during floods cannot be captured. Indeed, at least in the case of single event model runs, the surveyed starting morphology is one modified by recession processes, and so may not be representative of the morphology during the event peak.

14. Section 5.2.1. Following on from above, the discretisation of events into a single "time step" does not allow for the possibility that multiple scour and deposition cycles may occur within the actual event, with topography being "recycled" and the ECD capturing less the signature of discrete processes but a blurred composite. This issue is an old one of course (and befuddles application of the "morphologic method" for measuring bedload).

15. Section 5.2.3. I suggest a Discussion paragraph here around the straight ditch-cutting behaviour shown by the decadal simulation of the Feshie. It's an important concern for running the model for long periods, particularly since it turns an aggrading reach into a degrading one. This also compromises the statement on page 17, line 30-31 regarding unknown long-term computational stability – I'd say there is already a known issue appearing as it produces un-natural emergent behaviour.

Technical corrections/suggestions

P1, L20: What does multi-scalar mean here; in fact, why do you need to say this at all?

P7, L14: Supply exhaustion might affect the actual sediment transport but it doesn't change the transport capacity.

P8, L22: Qb is the unit bedload transport.

P15, L9: There's not much glacial melt in the Rees – it's mainly rainfall-driven with a seasonal snow and snowmelt signal.

P17, L11: These are validation results, not calibration results.

P 17, Equation 17: The union sign should appear in the denominator. As it is, the intersection sign appears in both numerator and denominator.

P19, L5: Section reference wrong – this is Sect 4.2.1.

P21, L6: Suggest replacing "model" with "models with the different PLDs".

P28, L10: Insert "a steady flow set to the event peak discharge and" between "using" and "path".

Table 1: It should be D84 in the 6th column, not D50 (as evident from lines 19-27 on page 6). Also, it should be C-W Roughness (after Colebrook-White) not W-C roughness.

Figure 10: The text is too small to read.

---

## Referee Comment (RC3) · Anonymous Referee #3 · 16 May 2018

Overview:

This paper presents a new morphodynamic modelling approach, whereby sediment transport is simulated through particle travel lengths algorithm rather than the more traditional flow-field and gravity-driven sediment algorithms. The model is applied to two case-study river reaches over different time scales.

Evaluation:

The paper is well-written and logically structured. The approach is novel and appears quite promising, both in terms of flexibility and initial results. I very much like the concept of the paper, and I would like to see it published eventually. However, the study could be significantly strengthened by a more comprehensive comparison with existing

modelling approaches, as outlined below. This additional element of analysis would probably constitute a major revision.

Comments:

1) The paper presents a new approach to simulating sediment transport in braided rivers. The authors claim advantages over existing simulation approaches such as reduced complexity modelling (RCM) which is lacking physical explanation and fidelity (p3 ln5) and computational fluid dynamics (CFD) which has too high computational overheads (p3 ln12). However, it is not clear from the paper that the new approach proposed by the authors produces comparable or better results than these existing approaches in terms of simulated morphologies. A direct comparing and contrasting with simulation results from these existing RCM and CFD approaches, highlighting both strengths and weaknesses of the particle travel lengths approach, would thus significantly strengthen the paper. The authors clearly are familiar with Delft-2D which should provide suitable CFD comparison simulations. On the RCM side, a model like CAESAR (with which at least one of the authors is also familiar) could provide suitable comparison simulations. The authors are of course welcome to choose other CFD and RCM models to compare to. But both these suggested models are able to simulate event-based scenarios and both are capable of simulating transport of multiple grainsizes, their simulations should be directly comparable to the simulations with the authors' particle travel lengths approach.

2) The authors note that the CFD models rely on the Exner equation to calculate bed elevation change (p3 ln10; p8 ln7). First it is worth noting that the RCM essentially do the same, in one way or another. More importantly, the authors suggest that they use an alternative approach to sediment continuity (p8 ln18). However, it is not clear that this indeed is the case. The authors present an approach to calculate the total erosion as a scour depth (p10 eq7). But surely, when it comes to adjusting the bed elevations the authors will still apply an equivalent of the Exner equation to ensure that the amount of material that is eroded matches this depth of erosion (or bed elevation
change).

3a) Lateral erosion is calculated in a simplified manner, i.e. scaled to near-bank shear stress (p9 ln18). In their conclusion the authors note that is unknown if this simplified lateral erosion model will provide stability over longer-term simulations. But it seems that their approach is similar, at least in its core concept, to the approach by Ikeda et al. (1981) that scales lateral erosion to near-bank excess flow velocity and that was later successfully applied in several studies over longer-term simulations (e.g. Howard, 1992, 1996; Sun et al., 1996, 2001; Stolum, 1998; and many others).

3b) It is not entirely clear how the lateral erosion module is implemented in practice. First all cells with steep slopes are identified, as possible targets for lateral erosion. For this steep slopes apparently are those with a gradient >7%. This seems excessively low, as most banks with gradients < 30% will be stable. The 7% threshold was identified through calibration (p9 ln25), but it is not clear how this calibration was done, to what accuracy, or on what data. Near-bank bed shear stresses are calculated using a 3x5 neighbourghood window, although it is not entirely clear how the orientation of the neighbourhood is determined. It seems to be based on the dominant cardinal aspect (fig 2.3A), but this is not explicitly identified as such. Finally, the total extent of the bank failure is calculated (p10 eq11). I presume this relates to the red delineated area in Fig 2.4, but it is not clear how that shape of that area is obtained – despite the authors attempt to describe this (p10 ln13). Further is not clear why there only are erosion values in the brown cells (Fig 2.4A) whilst these only are a sub-set of the red delineated bank erosion extent (Fig 2.4).

4) The authors lament the lack of physical explanation and morphological fidelity in RCM (p3 ln5), although they do not provide a proper argument or reference to support this claim. It is undoubtedly true that RCM, by their very nature, make some very simplifying, rule-based assumptions – but that does not necessarily mean that they therefore also lack physical explanation or morphological fidelity. Moreover, the authors make several very simplifying rule-based assumptions in their own approach – most

notably in the particle travel length approach itself, but also in the approach to sediment continuity, sediment deposition, and bank erosion. Thus, it could well be argued that the authors' model is itself a RCM (except for its CFD derivation of the flow field).

5) The model did not produce avulsions seen in the field (p22 ln13). Is the model inherently incapable of producing avulsions? Or is it capable in principle, but just did not do it in these simulations. If the former, this seems a major drawback (a fatal drawback??) for an algorithm that is designed specifically for braided rivers. If the latter, what would be the reason for not simulating the avulsions. This also relates to the broader discussion, where the authors claim that the model did reproduce all field-based braiding mechanisms (p28 ln29). This somewhat contradictory conclusion arises because the authors base this on a set of 10 braiding mechanisms (identified section 2.7.3). But, rather curiously, avulsion is not one of these braiding mechanisms. Subsequently, the authors claim that their model can simulate eight of these braiding mechanisms from sediment transport alone (p23 ln 7) and two more with additional algorithms. In other words, all 10 braiding mechanisms were reproduced in the simulations (p28 ln29). However, the key process of avulsion, although observed in the field (p22 ln13), is not considered in this – which seems rather flawed.

Minor edits:

p9 ln7: Eq.(10) –> Eq.(11)

p12 ln2: many –> may

fig 2: It is somewhat confusing that subfigure 3A is placed next to subfigure 2. Intuitively one would expect each of the detail views to be associated with the workflow view to the left of it. It is for 1, 3B and 4, but not for 3A.

fig 2: What is the colour scale for the figures in the third column?

fig 2: Caption needs adjusting to account for third column.

References:

Ikeda, Parker, Sawai (1981). J Fluid Mech 112, 363-377.

Howard (1992). in: Lowland Floodplain Rivers (Carling & Petts, eds), 1-41.

Howard (1996). in: Floodplain Processes (Anderson, Walling & Bates, eds), 15-62.

Stolum (1998). GSA Bull 110, 1485-1498.

Sun, Meakin, Jossang, Schwartz (1996). Wat Res Res 32, 2937-2954.

Sun, Meakin, Jossang (2001). Wat Res Res 37, 2243-2258.

---

## Author Comment (AC1) · 22 Jun 2018

General Comments

Referee: The manuscript presents an effort of modelling braided river morphodynamics by combining a 2D hydrodynamic model and a path-length based algorithm. The topic is relevant to an important issue on the earth surface dynamics, and it is should be interesting to readers of this journal. The aim of present hybrid approach is to develop a new framework of model to predict the braided river processes with limited computation time. Model predictions have been compared with two natural braided river for multi-scalar verification. The object of this study is clear. However, basic method is not

described sufficiently, and then it is difficult to estimate the value of the new model. In addition the prediction is not close enough to the measurement even in the statistical meaning, and the discrepancy is not explained enough. Therefore, in my opinion, the current version of this manuscript should be improved before being accepted by the journal for a publication.

Response: We thank the reviewer for their positive comments on the importance and relevance of this work, and for their insightful suggestions for revising the paper. We have made changes to the methodological descriptions (Section 2) within the manuscript in accordance with the Reviewer's suggestions, and believe that this section has been considerably improved clarified by this, and the subsequent two reviewers' comments.

Specific Comments

Referee: 1. Page 5, Lines 21-25, Section 2.1: It is not clear if time derivatives are included in the model? In the model description, "Delft3D solves the shallow-water form of the Navier-Stokes equations, which related changes in momentum (left-hand terms) in time and space . . .", while the terms of time derivative are not included in Equations 1-2.

Response: They are not; the Reynolds Averaged Navier Stokes Equations are used in Delft3D. This is a time-averaging of the Navier-Stokes Equations, and thus does not require time derivatives to be included. We've updated the text to reflect that the RANS are used here.

Referee: 2. Page 6, Lines 1 & 8-9: "Where x and y, respectively, denote the streamwise and cross-stream directions of velocity (u, v), . . ." Does the "streamwise" mean the river direction? Because the "For all modelling, we employed fixed Cartesian orthogonal grids", and the velocity changes over time and space. It is unclear whether the x and y is in a Cartesian coordinate system or other curvilinear coordinate system?

Response: Cartesian coordinates were used for all model simulations; we have specified that DEMs used in hydraulic modeling were rotated such that x-directions corresponded to the streamwise direction and y-directions corresponded to cross-stream directions.

Referee: 3. Page 8, Lines 15- 20: This part is the key solution of the proposed model framework, however, it cannot be understand clearly how to calculated the morphological change rate by using Equation 7 and with the concept of "integrated sediment transport pathways". More description and comments are needed at least here. Without a confirmed understanding, the value of the new model cannot be estimated sufficiently and the results would not be analyzed correctly.

Response:Thank you for pointing this out. The succinct answer is that an Exner/continuity approach for computing bed elevation change is suitable at short model timesteps, but not at the event-scale timesteps that MoRPHED is intended for. These long timesteps, in combination with Eq (6) would lead to topographic and numerical instabilities very quickly. In lieu of the traditional Exner-based approach, we used Eq. (7) to predict bed sediment scour, and in combination sediment derived via bank erosion (2.3), this material was transported downstream and deposited according to a user-specified path length distribution (Section 2.4).We have clarified how Eq. (7) was used in the model, and also why Eq. (6) wasn't used, and believe that this will more fully convey the operation of MoRPHED.

Referee: 4. Page 9, Lines 25-30: It is not clear, when a bank erosion would occurred. Is there any relationship between the "7%" and angle of repose? What is the mean of "we set this threshold area to 30 cells, again adjusting this value to mirror the size of field-observed bank erosion patch"? This part is a key procedure for the river migration, however the meanings are too ambiguous.

Response: The 7% threshold was determined by calibrating predicted areas of bank retreat, as a function of Equation 11, to field-observed areas known to have undergone

bank erosion and lateral migration. We've updated the text to reflect that while the 7% threshold used here was certainly below the angle of repose for unconsolidated gravel, we decided to take an inclusive approach to delineating candidate cells for bank erosion. The 30-cell area threshold was used to limit bank retreat to groups of cells exceeding a certain area for computational efficiency. We believe that the edited text will improve the clarity of this section.

Referee: 5. Page 10, Section 2.4: The general meaning is not clear for this section. Firstly, "velocity vectors need not pass through the centre of each cell." Why? And what is the result for the "Bed/Bank Sediment Transport and Deposition"? Secondly, how to combine the 5x5 deposition window of cells and the path length distribution in the model? Comments and figures should be used to illustrate the complex an core relationships.

Response: We've reworded the writing in this section to clarify that (1) while MoRPHED is a raster, or grid-based model, Delft3D produced streamlines which were inherently vector-based. To reconcile these two data types, streamlines were 'snapped' to the nearest cell centre for subsequent computation of sediment scour, transport, and path-length mediated deposition, and that (2) the result of this workflow is the deposition of a fraction of entrained sediment, which is given by the downstream distance from the entrainment location and the user-specified path-length distribution.

Referee: 6. Page 12, Line 5: How to determine the Z_ACT?

Response: In response to the suggestions of multiple reviewers, we have removed the original Section 2.6, which detailed the grain size and stratigraphic evolution component of the model, as this manuscript only presents results from single-fraction modeling runs. Subsequent sub-sections within Section 2 have been re-numbered accordingly.

Referee: 7. Page 13, Lines 25-30: DoD means "Difference of DEM" or "DEM of Difference" or others?

Response: DoD refers to a DEM-of-Difference; this is articulated in the Section heading and on line 6, page 14 of the revised manuscript.

Referee: 8. Page 18, Line 30: There is only a single sub-section 4.2.1 in this section.

Response: We have removed the text that referred to Section 4.2.1 within that same section.

Referee: 9. With regard to results and discussion, it is difficult to assess the agreement and disagreement of predictions without fully understanding of the model.

Response: We believe that the clarifications made to the methods and algorithms underlying the model (Section 2) will provide a better grounding for understanding the Results and Discussion components.

---

## Author Comment (AC2) · 22 Jun 2018

General Comments

Referee: Overview This is a well written exposition of a novel hybrid morphodynamic modelling approach that seeks to overcome the CPU-time limitations of conventional physics-based numerical models, thereby enabling the simulation of braided rivers over extended time and spatial scales. The CPU-time savings accrue from setting the morphological time step at the event scale. The event hydrograph is represented by a single discharge for which a robust steady-state hydrodynamic solution is derived. This is used to estimate entrainment from bank erosion and bed scour, with the entrained

sediment then advected downstream using an assumed distribution for particle travel length and also paper dispersed laterally. The approach is demonstrated using the case examples of the braided Rees and Feshie Rivers, with comparisons made between surveyed and modelled DEMs-of-difference in regard to volume-weighted elevation change distributions and classifications of braiding process. Nested "experiments" are also run that compare the model results when an event is represented by several quasi-steady flows rather than one, and when different types of path-length distributions are assumed. Extended discussion examines the limitations observed with the model results and around model assumptions, with good pointers on where to focus improvements.

Response: Thank you for these positive comments on the paper and its relevance to the field.

Specific Comments

Referee: 1. On page 7 lines 1-6 nothing is said about the duration of the steady upstream discharge (which is set at the peak discharge). Surely, this is critical for scaling the volume and at least the erosion depth axes on the ECDs, and for matching these to the surveyed ECDs. How was this set?

Response: This is because we did not attempt to scale sediment transport with event duration; the model presented here is purely event-based, operating at a timestep of a single flood, regardless of that flood's duration. Our aim was to assess the predictive capability of such an approach. We have added text to clarify this, along with specifying that floods of longer duration, specifically those capable of producing multiple entrainment events for particles, or which fundamentally alter bar spacing and path length distributions, may require multiple model timesteps. Further, Section 4.2.1. does provide an experiment where a single flood of long duration is discretized into multiple timesteps, thus demonstrating one approach for dealing with extended periods of competent flow. We have added text to Section 2 (specifically on pages 8 and

9) to emphasize this point.

Referee: 2. Pursuing this further, applying a scaling-factor to the bedload transport function would induce similar effects to varying the event "duration", so what about using this concept as a formal calibration approach – adjusting the duration or BT function to align model ECDs to the few observed field results, then running the so-calibrated model for longer times scales? It seems that so far, the paper only considers using the Rees and Feshie field data for validation purposes, not calibration.

Response: This is an interesting suggestion, and although a rigorous calibration using adjustment of the bedload transport function or event duration was not employed here, it may represent a promising way forward in improving model fidelity. Further, a simple scaling of the bedload transport function (Eq. 7) may elucidate the variability in scour depth on a system-to-system basis, providing insight into the propensity or resistance of various channel planforms to geomrophic change. While such a calibration exercise is beyond the scope of the present manuscript, we have alluded to the potential for future use of such an approach on Page 27 (Discussion).

Referee: 3. On page 7, around lines 20-25, it appears that only the median grain size is used for calculating the threshold stress, which is then used to calculate the bed-load transport with equation 8 (for use in equation 7). While the method is claimed to be multifractional, there is nothing said about hiding factors, partial transport, different pathlength distributions by grainsize, etc.). Equal-mobility is implicitly assumed, so effectively, entrainment and deposition are undertaken only considering the median size, so (apart from remixing of the active and sub-surface layers at the end of the event, as explained on page 30) the method doesn't actually do multi-fraction processes. This means that it will not produce armouring, which can be an important control on scour depth. So, claiming the approach is multi-fraction is not true – it only pays lip-service to being multi-fraction.

Response: Because this manuscript only presents the results of single-fraction modeling, as the reviewer rightly points out, we've removed this section, and the original Figure 4, for simplicity.

Referee: 4. On page 8, line 2 -3, it's not clear if 10 cells are being taken upstream and another 10 cells downstream or if 10 cells are being taken centred on the current cell. If the former, then in the case of the Rees mode,l 40 m averaging appears too excessive. Please clarify and justify if need be.

Response: It is the former; we have added text to clarify this. While smaller values for this averaging may be suitable on the Rees, we found that this approach avoided mor­phologic artifacts (e.g., scouring excessively deep pools) during model runs. We also believe this averaging distance is justified given that the confluence-diffluence spacing on the Rees, and hence the average distance between areas experiencing scour and deposition, is ~350 m (Figure 3), which is approximately 9 times the averaging window used here.

Referee: 5. On page 8, lines 15-18, the logic around the alternative to the Exner equation is never completed. What appears to be done (but is not properly explained) is that "clearwater" scour is first calculated everywhere it can potentially occur, with a Ds assigned to each cell, then a deposited sediment thickness is assigned to cells downstream of erosion sites according to the path-length distribution and smearing algorithms. Thus, some cells will only experience scour, some only deposition, and some that scour will also experience deposition from sediment sources from scour sites upstream. Net elevation change at-a-cell is therefore calculated as the difference between scour and deposition. This needs to be explained, and if I have the story wrong, the right story needs explaining please.

Response: This interpretation is completely correct; we have further clarified these process representations on page 9.

Referee: 6. On pages 10-11, a diagram (similar in concept to Fig 2) is needed to help explain the distribution of sediment deposition.

Response: We've added a new figure (revised Figure 4) that explains the algorithm for sediment distribution.

Referee: 7. Page 14, lines 21-41, the methodology used for classifying braiding mechanisms is not stated. Was this purely subjective and done manually, or were algorithms employed that obeyed a set of rules? What were the guiding rules? Was the same approach applied to the field data and the model output?

Response: This was completed for both field and model data. The approach is subjective and interpretive, similar to geomorphic mapping of surficial deposits. However, the approach has been widely used, and efforts aimed at automating the process have been introduced in the literature (see, for example, Kasprak et al. 2017, ESPL). At the same time, divergent braiding mechanisms are revealed quite well in DoDs (for example, high-magnitude narrow swaths of erosion are indicative of bank erosion, while thin-mantled widespread deposition indicate that overbank sheets were responsible for elevation changes in a given area).

Referee: 8. Page 16, lines 6-9. Isn't it more a case that the Rees, by virtue of its frequent runoff events and significantly finer bed material (from its schist catchment), is a lot more labile which inhibits woody vegetation (and associated fine sediment trapping) from establishing?

Response: This is likely the case, although it's unclear whether vegetation inhibits geomorphic change or vice versa; we've inserted text near here to note that the dynamism of the Rees likely precludes vegetation establishment.

Referee: 9. Section 4.1.2 addresses only validation results, since no calibration was actually done. So, in line 11 on page 17 replace "calibration" with "validation". Also, on page 17 around lines 19-22 it would be useful to mention the Ddiff and Fc numbers given in the Figure. Also, I recommend also providing the RMSE of depth differences. The mean Ddiff statistic reported only informs on any overall bias in depth and tells nothing about how badly the depth may have been predicted locally.

Response: We've changed 'calibration' to 'validation' throughout this section, and have stated the Ddiff and Fc values in the text as well as in the associated Figure (6), along with listing the RMS of depth differences in Figure 6.

Referee: 10. Section 4.2, page 17-18, several things. First, some more information is needed here to clarify what was done in the modelling – particularly confirm what discharge was used and how long it was run for. Second, no mention is made at all in the text about thresholding but it is the thresholded results of volume change that are being discussed, not the gross ones. I would expect to see some comparison and discussion, particularly since the modelling doesn't have thresholding issues. Third, in lines 22-23 the statement that "average magnitudes of erosion and deposition agree well between field and model results" is a bit rose-tinted, because the differences appear to be in the range of 33-50%. Fourth, it would be very informative to compare the MoPHED modelled results for the Rees event with the results of the full Delft3D model run of the same event (already done by Williams et al). Fifth (and again), what were the durations of the steady-flow model runs, how were they decided, and how do they impact on the results (I am assuming that when "model run time" is mentioned that this is processing time, not event duration). It would be useful to show the discretised quasi-steady hydrograph time-spans for the single event discretisation and for the 3-event discretisation on top of the hydrograph in Fig 7A.

Response: In Section 4.2, we've now clarified that a single discharge of 75 cumecs was used in this modeling; there was no duration component to this value (see item 1 above); this also pertains to the 5th suggestion provided by the reviewer here. There is no 'duration' for an event modeling run (that is, modeling was in no way scaled by the duration of the event, but instead can be thought of as a single 'snapshot' in time. We have also indicated that the thresholded model results are being presented here; the rationale for using the thresholded results was simply to allow for consistent comparisons between field and model DoDs. See also our response to Reviewer 3, but in short we've added a new section in the Discussion (5.2.3) that directly compares

**ESurfD**
our results with those of Williams et al. (2016, WRR). We've removed the sentence about agreement between field and model average depths of erosion and deposition, given that the reviewer rightfully points out the discrepancy between these values, and the fact that this sentence interpreted results and was out of place in this Section.

Referee: 11. Page 19, lines 20-29. Isn't all this just saying that you picked events when the flows were competent as indicated by Ashworth and Ferguson's observed threshold? The "low bankfull" flow story seems like an unnecessary complication.

Response: We've eliminated the references to 'low bankfull' and have simply noted that these events were competent.

Referee: 12. Page 22, line 10. I disagree with this stated general agreement in form of the ECD. The model-predicted ECD is clearly more erosion skewed than what was observed. Indeed, what the model appears to have done with these prolonged runs is to cut straight, deepening "ditches". I've experienced this before with long Delft3D runs (e.g. Singh et al), and I suspect it stems from inadequate handling of bank erosion and probably also lack of armouring. The model has also produced net degradation that is 2+ times the magnitude of the net aggradation that was observed in the field. So, there is an un-natural emergent behaviour occurring here in the model, which needs to be discussed – and hopefully dealt to with some clever workaround. It would be useful to provide the changes in the sinuosity over the decade of model runs to quantify this trait.

Response: We have modified the sentence regarding agreement between field and model ECDs to only refer to the depositional fraction, which was similar between both field and model, and have discussed this and referenced Singh et al. (2017)'s work using Delft 3D on page 22. The sinuosity (and change therein) is presented in Figure 11, but exhibited little variability following the model run, likely because many of the anabranches in 2003 did not completely fill in (as a result of the lack of avulsions), but rather remained hydrologically active in addition to the deepened central anabranch.

However, the number of channel nodes provide a good indication that after 10 years, the channel planform was simplified, and this reduction is presented in the text along with Figure 11.

Referee: 13. Page 23, line 25-26. Yes, this is an important point – the latency between flow, sediment transport and morphologic response is not captured by the quasi-steady approximation, so transient features during floods cannot be captured. Indeed, at least in the case of single event model runs, the surveyed starting morphology is one modified by recession processes, and so may not be representative of the morphology during the event peak.

Response: Thank you for pointing out the importance of this statement.

Referee: 14. Section 5.2.1. Following on from above, the discretisation of events into a single "time step" does not allow for the possibility that multiple scour and deposition cycles may occur within the actual event, with topography being "recycled" and the ECD capturing less the signature of discrete processes but a blurred composite. This issue is an old one of course (and befuddles application of the "morphologic method" for measuring bedload).

Response: This is a good point, and the text we've added to address comment (1) above mentions that this is a particular disadvantage for using an event timestep to model long-duration floods.

Referee: 15. Section 5.2.3. I suggest a Discussion paragraph here around the straight ditchcutting behaviour shown by the decadal simulation of the Feshie. It's an important concern for running the model for long periods, particularly since it turns an aggrading reach into a degrading one. This also compromises the statement on page 17, line 30-31 regarding unknown long-term computational stability – I'd say there is already a known issue appearing as it produces un-natural emergent behaviour.

Response: We have added several sentences on this behaviour on page 28, incorporating the reviewer's suggestion that this over-scouring may be an artefact of a lack of bed armouring, improper bank erosion representation, or a combination thereof.

Technical Corrections

Referee: P1, L20: What does multi-scalar mean here; in fact, why do you need to say this at all? Response: We've removed 'multi-scalar' from the abstract text.

Referee: P7, L14: Supply exhaustion might affect the actual sediment transport but it doesn't change the transport capacity.

Response: Removed 'capacity' from this sentence.

Referee: P8, L22: Qb is the unit bedload transport.

Response: Augmented the text here to note that Qb is referring to the unit bedload transport rate.

Referee: P15, L9: There's not much glacial melt in the Rees – it's mainly rainfall-driven with a seasonal snow and snowmelt signal.

Response: Corrected to note that floods are the byproduct of snowmelt and rainstorms.

Referee: P17, L11: These are validation results, not calibration results.

Response: Changed 'calibration' to 'validation' here.

Referee: P 17, Equation 17: The union sign should appear in the denominator. As it is, the intersection sign appears in both numerator and denominator.

Response: Thanks for catching this! Replaced the intersection with union sign in the denominator of Equation 17.

Referee: P19, L5: Section reference wrong – this is Sect 4.2.1.

Response: As in response to Reviewer 1's comment, we've removed this section reference.

Referee: P21, L6: Suggest replacing "model" with "models with the different PLDs".

Response: We've changed this to "modelling different path length distributions".

Referee: P28, L10: Insert "a steady flow set to the event peak discharge and" between "using" and "path".

Response: This text has been inserted as suggested.

Referee: Table 1: It should be D84 in the 6th column, not D50 (as evident from lines 19-27 on page 6). Also, it should be C-W Roughness (after Colebrook-White) not W-C roughness.

Response: We've changed both of these column headings as suggested.

Referee: Figure 10: The text is too small to read.

Response: Thanks for pointing this out – we've enlarged the text accordingly

---

## Author Comment (AC3) · 22 Jun 2018

General Comments

Referee: Overview: This paper presents a new morphodynamic modelling approach, whereby sediment transport is simulated through particle travel lengths algorithm rather than the more traditional flow-field and gravity-driven sediment algorithms. The model is applied to two case-study river reaches over different time scales.

Referee: Evaluation: The paper is well-written and logically structured. The approach is novel and appears quite promising, both in terms of flexibility and initial results. I very much like the concept of the paper, and I would like to see it published eventually.

[Figure]

However, the study could be significantly strengthened by a more comprehensive comparison with existing modelling approaches, as outlined below. This additional element of analysis would probably constitute a major revision.

Response: We thank the reviewer for these generally positive comments, and have addressed their suggestions in the pages that follow.

Specific Comments

Referee: 1) The paper presents a new approach to simulating sediment transport in braided rivers. The authors claim advantages over existing simulation approaches such as reduced complexity modelling (RCM) which is lacking physical explanation and fidelity (p3 ln5) and computational fluid dynamics (CFD) which has too high computational overheads (p3 ln12). However, it is not clear from the paper that the new approach proposed by the authors produces comparable or better results than these existing approaches in terms of simulated morphologies. A direct comparing and contrasting with simulation results from these existing RCM and CFD approaches, highlighting both strengths and weaknesses of the particle travel lengths approach, would thus significantly strengthen the paper. The authors clearly are familiar with Delft-2D which should provide suitable CFD comparison simulations. On the RCM side, a model like CAESAR (with which at least one of the authors is also familiar) could provide suitable comparison simulations. The authors are of course welcome to choose other CFD and RCM models to compare to. But both these suggested models are able to simulate event-based scenarios and both are capable of simulating transport of multiple grainsizes, their simulations should be directly comparable to the simulations with the authors' particle travel lengths approach.

Response: We appreciate this comment and agree that such a benchmarking study would be both interesting and insightful with regard to the path-length model's utility and validity going forward. At the same time, we feel that such an endeavor is well beyond the scope of the current manuscript, which simply seeks to present and

provide case studies for the use of a novel modeling approach, one which we believe makes significant strides in computational efficiency by moving beyond the traditional particle-tracking and/or Exner-based approaches for computing morphodynamics. Such a benchmarking approach would be a logical next step in our development of this model and would make for an interesting follow-on manuscript; while not part of the current work, we have added several pages to the Discussion within a new section (5.2.3), which take advantage of past RC/CA and CFD modeling done by Williams et al. (2016a,b) on the Rees River to directly compare the results of all three modeling approaches and suggest improvements common among the techniques going forward.

Referee: 2) The authors note that the CFD models rely on the Exner equation to calculate bed elevation change (p3 ln10; p8 ln7). First it is worth noting that the RCM essentially do the same, in one way or another. More importantly, the authors suggest that they use an alternative approach to sediment continuity (p8 ln18). However, it is not clear that this indeed is the case. The authors present an approach to calculate the total erosion as a scour depth (p10 eq7). But surely, when it comes to adjusting the bed elevations the authors will still apply an equivalent of the Exner equation to ensure that the amount of material that is eroded matches this depth of erosion (or bed elevation change).

Response: This is correct, and was a question that was also raised by Reviewer 2 above, and we have elaborated on the approach used on page 8, along with adding a new Figure (Fig. 4) to illustrate the erosion and deposition algorithms used in the model.

Referee: 3a) Lateral erosion is calculated in a simplified manner, i.e. scaled to near-bank shear stress (p9 ln18). In their conclusion the authors note that is unknown if this simplified lateral erosion model will provide stability over longer-term simulations. But it seems that their approach is similar, at least in its core concept, to the approach by Ikeda et al. (1981) that scales lateral erosion to near-bank excess flow velocity and that was later successfully applied in several studies over longer-term simulations (e.g.

Howard, 1992, 1996; Sun et al., 1996, 2001; Stolum, 1998; and many others).

Response: We appreciate the Reviewer noting the similarity between our use of near-bank shear stress to predict bank retreat with those approaches developed by previous researchers. We have included the references listed above on page 11. When we raised the question of whether such an approach would yield stability over longer-term simulations, we were largely referring to the use of a once-per-flood (i.e., event-scale) technique for eroding bank material, as opposed to computing lateral retreat many times over the course of a flood. We have clarified this on page 29.

Referee: 3b) It is not entirely clear how the lateral erosion module is implemented in practice. First all cells with steep slopes are identified, as possible targets for lateral erosion. For this steep slopes apparently are those with a gradient >7%. This seems excessively low, as most banks with gradients < 30% will be stable. The 7% threshold was identified through calibration (p9 ln25), but it is not clear how this calibration was done, to what accuracy, or on what data. Near-bank bed shear stresses are calculated using a 3x5 neighbourghood window, although it is not entirely clear how the orientation of the neighbourhood is determined. It seems to be based on the dominant cardinal aspect (fig 2.3A), but this is not explicitly identified as such. Finally, the total extent of the bank failure is calculated (p10 eq11). I presume this relates to the red delineated area in Fig 2.4, but it is not clear how that shape of that area is obtained – despite the authors attempt to describe this (p10 ln13). Further is not clear why there only are erosion values in the brown cells (Fig 2.4A) whilst these only are a sub-set of the red delineated bank erosion extent (Fig 2.4).

Response: The text within Section 2.3 has been significantly revised in accordance with Reviewer 2's suggestions above. We have noted in the revised text that while the 7% threshold is certainly below the angle of repose, it provides an inclusive first-approximation of cells that may be candidates for lateral retreat, and these cells are further refined through the use of a near-bank shear cutoff. In addition, because lateral retreat was only computed once per flood, we took an inclusive approach (i.e., low

slope threshold) here to account for those cells that may undergo slope failure from progressive bank steepening over the course of a flood event. The 7% threshold was initially chosen by examining the average slope of cells that subsequently underwent lateral retreat in field data, which has been noted on page 10. The 3x5 window was indeed oriented in the cardinal aspect of the candidate cell, and we have now noted this in the text on page 10. The values within the brown cells (Panel 4A, Figure 4) correspond to the computed extents around those cells where bank erosion occurs – essentially, the brown cells are "padded" by this value, and the cells falling within that padding window undergo bank erosion, which produces the red polygon in Panel 4. We have clarified this approach on page 11.

Referee: 4) The authors lament the lack of physical explanation and morphological fidelity in RCM (p3 ln5), although they do not provide a proper argument or reference to support this claim. It is undoubtedly true that RCM, by their very nature, make some very simplifying, rule-based assumptions – but that does not necessarily mean that they therefore also lack physical explanation or morphological fidelity. Moreover, the authors make several very simplifying rule-based assumptions in their own approach – most notably in the particle travel length approach itself, but also in the approach to sediment continuity, sediment deposition, and bank erosion. Thus, it could well be argued that the authors' model is itself a RCM (except for its CFD derivation of the flow field).

Response: This is a fair point, and we have restricted our statement on RCMs to simply note that one shortcoming of these models is that they are often unsuitable for prediction of any specific geomorphic system (in the sense that the Murray and Paola model produced a characteristic planform of 'a' braided river, but not a specific braided river), and are thus useful for investigating the effects of shifting boundary conditions in a generalized sense, but not for explicit predictions of channel dynamics for any one system.

Referee: 5) The model did not produce avulsions seen in the field (p22 ln13). Is the

model inherently incapable of producing avulsions? Or is it capable in principle, but just did not do it in these simulations. If the former, this seems a major drawback (a fatal drawback??) for an algorithm that is designed specifically for braided rivers. If the latter, what would be the reason for not simulating the avulsions. This also relates to the broader discussion, where the authors claim that the model did reproduce all field-based braiding mechanisms (p28 ln29). This somewhat contradictory conclusion arises because the authors base this on a set of 10 braiding mechanisms (identified section 2.7.3). But, rather curiously, avulsion is not one of these braiding mechanisms. Subsequently, the authors claim that their model can simulate eight of these braiding mechanisms from sediment transport alone (p23 ln 7) and two more with additional algorithms. In other words, all 10 braiding mechanisms were reproduced in the simulations (p28 ln29). However, the key process of avulsion, although observed in the field (p22 ln13), is not considered in this – which seems rather flawed.

Response: The model did produce avulsions, the most notable of which is the development of an incised anabranch on the left-hand side of the braidplain at the upstream end of the reach (Figure 11), where no channel existed prior to the simulation. On page 22, we were referring to the fact that the model did not produce avulsions in the same locations as observed in the field, and overall, the model produced a more stabilized (i.e., incised) channel planform than that seen in the field, which implies that the model did not produce avulsions at the rate seen during the period 2003-2013 on the Feshie; this is further evidenced by the reduction in channel nodes in the model as compared to field data. Algorithmically, the model is indeed capable of producing avulsions through the lateral retreat of banks (and direct downcutting of braidplain/bed areas when flow is sufficiently high). We have augmented the text on page 22 for clarity.

We used the braiding mechanisms developed by Ashmore (1992) and expanded by Wheaton et al. (2013), avulsion was not explicitly included in the list of ten mechanisms assessed here. We would argue, however, that avulsions are the product of one or more of the braiding mechanisms we did assess. In particular, bank erosion, chute

cutoff, and central bar development, among others, can lead to avulsion (e.g., Ferguson, 1993). In our simulations, it is possible that these processes did not occur with sufficient magnitude relative to channel incision such that the frequency of avulsions seen in the field was matched by that in the model. We have added a paragraph to the Discussion (page 24) to emphasize this.

Technical Corrections

Referee: p9 ln7: Eq.(10) –> Eq.(11)

Response: We're unsure what correction the reviewer is suggesting here; we've checked the numbering and references to each equation and can't find any mistakes.

Referee: p12 ln2: many –> may

Response: This section (originally 2.6) has been omitted from the text in the revised manuscript.

Referee: fig 2: It is somewhat confusing that subfigure 3A is placed next to subfigure 2. Intuitively one would expect each of the detail views to be associated with the workflow view to the left of it. It is for 1, 3B and 4, but not for 3A.

Response: We agree, and have added arrows linking the sub-figures with their appropriate panels in Figure 3 to avoid confusion.

Referee: fig 2: What is the colour scale for the figures in the third column?

Response: We believe that the reviewer is referring to Figures 7, 9, and 11 here. The colors correspond to each of the braiding mechanisms, which are described in subpart (G) of each figure. We've updated the caption to reflect this.

Referee: fig 2: Caption needs adjusting to account for third column.

Response: See above

**ESurfD**

Interactive
comment

---

## Referee Report (RR1)

**Re-review of paper "Modelling braided river morphodynamics using a particle length framework" submitted by Kasprak et al.**

This is a re-review of this paper so I will cut to the chase. The authors have done a generally good job of attending to previous review comments but there remain several issues with the present version that should be addressed. I will discuss those first, then list smaller/minor page by page things warranting corrections.

1. In the abstract (and later in the paper) the notion of representing hydrographs as a series of steady states is misleading – it continues to imply a duration associated with each steady state (note that I fully understand how time is removed from the calculations), and it implies that this expediency is at the root of the computational time saving over CFD type approaches. The implication is that by chopping event hydrographs down into smaller and smaller blocks the approach might converge on a CFD result. But that is never done in the paper – the example provided from the Rees is actually three events back to back – it is not a single event hydrograph split in 3 time blocks.
   Indeed, a key (unstated) assumption of the approach is that each event modelled has adequate duration to allow the bedload to get to its destination – so you can see that breaking events up into smaller and smaller intervals will ultimately badly overcook the results, not give them greater precision. While I appreciate that this is never actually done, the paper hints that it could be.

2. On page 9 there are two things that niggle me. The first is that Montgomery's event averaged relation (eqn 7) is applied to the event peak discharge then assumed to be representative of the whole event – so all of sudden event averaged Qb becomes Qb at peak discharge, and ditto for bedload advection speed. What Eqn 7 is really predicting at the event peak is the active layer thickness. The second is that Eqn 8 is not the MPM formula – the scaling coefficient (8) is missing as are the density terms. The net result is that a factor of ~ 0.4 is missing from the equation. Hopefully this was an omission in the text and not the model.

3. Section 2.5. The input boundary condition option used needs to specified for each model run. What was done in the cases of the first event in a series or for single events, like the first Rees example – since there is not a preceding event calculation?
   Also,what was done at the lateral boundaries? It reads like bedload was allowed to diffuse across them but was not returned (e.g. on the opposite side) so the modelling should progressively lose sediment mass.

4. Two things on page 19. First, on line 18, the notion of altering the model timestep is misleading (as discussed in 1. above). The example used is actually a multipeak event, but is dominated by the big middle one. Note on line 20, 254 should be 259 (as elsewhere).

5. At the top of page 21, again, the assumption needs to be stated that the undisclosed event duration provides adequate time for the bedload to get to its destination. It would be a good idea to calculate this as an assumption check.

6. On page 22, there are potential implications of the upstream BC that require discussion. Around about line 12 it is important to say which upstream boundary condition is imposed. Also, I would like to see some discussion around how the sequencing of events impacts the results – for example, a large event before a small one will deliver a large wad of bedload into the top of the reach for the small event - promoting deposition there. It is mentioned that the top 25 m of the reach is not analysed but that is because of the hydraulic boundary condition issue. Surely the sequencing of events will affect the net apparent change

between the first and last event in the Feshie sequence? I suggest a plot of mean bed level vs time to explore this?

7. Section 5.1, which discusses emergent behaviour, really needs to discuss the emergent scour behaviour that develops. Given only a relatively small number of iterations (i.e. 185 events in the period 2003-2013), the tendency to scour, degrade, and simplify the channel network is quite notable (e.g. -8.07 elevation fall is very severe and quite unrealistic). While this is noted on page 22 (around line 25) with reference to the Singh et al Delft3D run, it doesn't mention that the Singh et al run also produced unrealistic results. So this issue appears to be being swept under the carpet whereas it really shows that the model in its present form is not to be trusted for predicting decadal or longer scale morphological evolution. While this is discussed some more on page 30, and explanations are offered, it seems pretty clear that the main channel has gotten into a (scour, increased flow capture, increased shear stress, increased ) feedback loop.

Page/by page comments:

P5, L 26: Need to say that Delft3D was run in 2D depth-averaged mode. Also, I suggest shifting the sentence on lines 9-12 of P6 to here.

P7, L1-4: Needs some edits. Also, give units of m to the Ks values.

P7, L11-12: Need to say at event timescales rather than coarser time intervals. The model is never being run at "coarser time intervals". The events are assumed to have undetermined duration. See also comment 1 above.

P7, L26-27: Sorry, but I've never heard of a braided river being exhausted of bedload through a hydrograph. This might be reasonable for suspended load. I suspect latency issues created by the upstream bedload feed boundary condition is something better to evaluate (see 6 above).

P8, L20: Needs to read Grad Qs. But also, why talk about Qs here then jump across to using Qb in Eqns 7 and 8?

P9, L 7: Need to say that Qb is the average unit mass-based bedload during the event.

P 9, eqn 8: give the dimensionally correct version of the MP-M formula.

P13, L17: Mode of what?

P13, L20: Suggestion only, but I struggle to see that $S_T$ is particularly informative here, as it really weights the average sinuosity of individual channels (which is a useful measure of the morphological evolution) by the braiding index. In other words, if you use $S_T$ you don't know if it's due to a sinuosity change or a BI change.

P13, L28: It's not large differences in the metric, it's large non-zero values (and say what -values indicate and what +values indicate).

P15, L24: Say 0.5 m grid models – otherwise it suggests models accurate to 0.5 m in elevation.

P16, L31: The model roughness parameter is the Nikuradse roughness length, ks.

P17, all of section 4.1.1: This is confusing and needs to be written. On P 7 you say you calculate ks off grainsize, then here you first suggest you derived this by trial error (i.e. by calibration),

then finally you say you just used Williams et al's calibration result. Is it that you used an initial grainsize-based value of ks, saw no reason to change this during calibration runs, and this aligned with what Williams et al got?

P 20, L4: I'm curious – how much of the bed was mobile at 20 m3/s? Was that a good estimate based on the more detailed results coming from the model (which evaluates Tc everywhere)?

P25, L6-8: It seems hardly surprising that the big 259 m3/s event dominated compared to the "pups" that came before and after it – particularly considering the threshold of motion influence. So I see no reason for the speculation that follows. I suggest deleting.

P27, L22: I disagree that it replicates the magnitude, rather, it produces similar.

P27, L 24: It would be helpful to say if Williams et al thresholded their model results as done in this paper – so that that apples are being compared with apples.

P32: I suggest acknowledging the referees!

---

## Author Response (AR2)

**FROM REVIEWER #2**

1) The strangest element of the model is the assumption that event erosion is independent of event duration (p10 ln1-3). Imagine two scenarios: scenario A is a three-day flood of 200 m3/s; scenario B consist of three one-day flood events of the same discharge, but separated by a short period of low flow. In the authors' approach, scenario B will result in three times as much erosion as scenario A. This seems counter-intuitive.

Nonetheless, the authors obtain seemingly reasonable results for their single event simulations on the River Rees and multiple event simulations on the River Feshie. This is especially notable since the authors "did not attempt to scale or correct estimates of bed sediment erosion as a function of flood duration" (p10 ln3). However, their method does introduce a dependency on event discretization or the choice of event magnitude threshold, discretizing multiple sub-events for the same simulations would simply result in more erosion and more geomorphic change.

Indeed, with their discretized hydrograph simulations the authors observe more erosion over a wider area than with the single discharge simulation (p19 ln21-24). This is not surprising, since the discretized hydrograph effectively consists of three "events" whereas the single discharge representation consists of one "event". Since event erosion is independent of event duration (p10 ln1-3), it is to be expected that adding two "events" as part of the discretization will result in more erosion. As the added events are of smaller magnitude (representing the rising and falling limbs of the hydrograph) there is no tripling of the erosion. A further discretization, representing more points on the hydrograph would result in more erosion still.

Can the authors elaborate on the further implications for the seemingly odd assumption that event erosion is independent of event duration?

*We have added a paragraph to the Discussion (Section 5.2.3) that deals with this assumption. To be sure, this is a simplified way of treating event-scale scour, and the Reviewer is right to point this out. At the same time, employing such an approach allows two advances to be made. First, we argue that the geomorphic community's understanding of event-scale deposition (i.e., path lengths and their relationship to geomorphic unit spacing) is robust, and similarly, computing sediment scour required to drive that depositional component at the event scale provides a practical way to employ what investigators have learned from developing path length relationships in the field and flume.*

*Secondly, treating erosion and deposition at the event scale is analogous to the vast majority of field surveys completed by fluvial geomorphologists; that is, most of the topographic surveys used to quantify fluvial morphodynamics are captured before and after a flood, and this modelling approach provides a way to square our field surveys, which are inherently at the event timestep, with a computationally-efficient method for modelling morphodynamics. We've noted in the added text that our approach only stands to benefit from (a) improved event-scale predictions of scour depth, and (b) a refined understanding of the role of flood magnitude/duration on sediment scour depth.*

2) The authors' bank erosion mechanism is not fully clear, and could be descried and illustrated in more detail.

2a)The authors state that "the model calculates the slope of all cells in the model domain […]. Cells that exceed a user-defined slope criterion […] are then identified as candidate cells for undergoing bank erosion" (p10 ln14-20). As this occurs for all cells in the model domain, this seems to suggest that bank erosion can occur on any cell of the model domain, i.e. not just on the banks, but potentially also on the

river bed or on the floodplain. Is there any selecting that the bank cells have to be dry and be bordering a wet cell?

*This is a good observation; to be candidates for bank erosion, cells do not necessarily need to be on the wet/dry boundary, but simply need to be steeply-sloping and have high adjacent shear stress. Inundated cells may therefore undergo bank erosion. This was included in the model because at the event peak (when hydraulic simulations were performed), steeply-sloping cells that may be riverbanks at lower discharges would likely be inundated, and using this approach, they are also candidates for erosion. We have included text clarifying this at the end of Section 2.3.*

2b) The bank erosion algorithm introduces a couple of seemingly arbitrary choices: erosion area threshold (30 cells), shear stress sampling area (3x5 window), average shear stress exclusion threshold (50 N/m2). Has there been any sensitivity analysis to investigate the impact of these arbitrary value choices? For the erosion area threshold the authors indicate that 30 cells produced good agreement with observed erosion patches (p10 ln28-29). Are the other choices equally calibrated to the observed field-data? Or are they based on other considerations? Would any deviation from these seemingly arbitrary values result in notably different results, or in numerical instability? Are these values likely to need adjusting for each River?

2c) Equation 11, determines the number of cells from which material will be removed during bank erosion? What is the basis for this equation? How was it derived? Why is shear stress divided by 3? Why is slope divided by 15? Is the division by 15realted to the size of the shear stress window (3x5)? If so, does it matter that not all cells in that window are necessarily included in the shear stress calculation.

*During model development, a number of iterations of Equation 11 were examined, and the values therein were adjusted to obtain the best possible agreement with areas of bank erosion delineated from successive DEMs on both the Feshie and the Rees Rivers. We did not conduct a formal sensitivity analysis or calibration approach, but rather examined the influence of varying those parameters until maximal agreement between field and model results was reached. To be sure, the precise values in Equation 11 are not physically-based in the traditional sense, but rather were arrived at via qualitative calibration.*

*More broadly however, and particularly in the Introduction, we have emphasized that the overarching aim of this study was to develop a highly simplified, and thus computationally-efficient, approach for modeling event-scale morphodynamics. To assess this model in a reductionist framework and compare our approach with CFD-like force balance methods is, practically speaking, to compare apples to oranges. This model blends aspects of both CFD modeling (i.e., hydraulic simulations) and reduced-complexity approaches (i.e., sediment scour and deposition, and bank erosion). Numerous other braided river morphodynamic models rely on similarly simplified rulesets for computing morphodynamics (Murray and Paola's 1994 model is a prime example), rather than a true physically-based force balance approach. The values used in Equation 11 are simply reflective of the best qualitative correspondence between modelled and field-derived estimates of areas undergoing bank retreat.*

*Identical values in Equation 11 were used for both the Feshie and the Rees, which have considerably varying grain sizes, vegetation, and amounts of fine sediment, and thus we believe the values in Equation 11 are fairly robust and applicable to a range of river systems. That being said, these values will likely need to be adjusted for rivers with exceptionally high degrees of bank cohesion; for example, river reaches that are densely covered by vegetation and/or contain large volumes of cohesive clays or muds. We have inserted text to this effect in Section 2.3.*

2d) It is not clear how the red area (Fig 2.4) is derived, nor is it fully clear how the volume of erosion is determined. The number of cells in the red area, i.e. n, is determined using equation 11. But I do not understand how the location of these cells is determined.

I raised the lack of clarity on this issue in my initial review. In their response the authors note that " the values within the brown cells (Panel 4A, Figure 4 [presumably the authors meant Fig 2]) correspond to the computed extents around those cells where bank erosion occurs – essentially, the brown cells are 'padded' by this value, and the cells falling within that padding window undergo bank erosion, which produces the red polygon in Panel 4. We have clarified this approach on page 11."

I do not think the authors' clarification on page 11 is very clear, nor is Fig 2. First of all, the title for fig 2.4A suggests that the values given are "erosion values", not "the computed extents around those cells where bank erosion occurs". Further, the text indicates that "material is removed from cells working in a direction opposite the bank cells' aspect" (p11 ln13), but in that case it is not clear how the red area can extend in two directions beyond the brown cells that river bank make up the end points of bank erosion patch (fig 2.4). Also it is not clear from which cells in the brown patch one would start to identify the n red cells. Presumably I would get a different cluster of n red cells if I start allocating them from the upstream end than if I were to start from the downstream end.

*The reviewer is correct regarding the interpretation of Figure 4A, which depicts the number of adjacent cells that will undergo bank erosion (working in a direction opposite to the initial cell's aspect). This value is given by Equation 11 in the text. We have modified the title of Figure 4 to clarify this.*

*The red polygon in Figure 2.4 depicts the total extent of bank erosion that would occur given the number of 3 x 3 neighbourhoods shown in Figure 2.4A (zoom-in) extending from each initial candidate cell for bank erosion. The direction in which that neighbourhood extends is completely a function of the initial cell's aspect direction, and thus the red neighbourhood can extend in any of the four cardinal directions away from that initial candidate cell.*

*The order (i.e., which brown cell one begins this routine at) is irrelevant, as the delineation simply draws 3 x 3 neighbourhoods around the brown cells, and the routine will produce an identical area of red cells in Figure 4 regardless of the order in which those cells are delineated.*

*We have modified the original text intended to clarify this routine, and we hope these modifications have made the bank erosion approach more understandable.*

Finally, material is removed from the n cells in the red area. The text states that "eroded cells are reduced in elevation to a level equal to that of the lowest cell in the initial neighbourhood surrounding the bank cell". Presumable this is a 3x3 neigbourhood, although this is not specified. Moreover, it is not clear if this refers to the neighbourhood of the eroding bank cell (i.e. red cell, fig 2.4) or the neighbourhood of the original bank cell (i.e. brown cell).

*The reviewer is correct that cells are lowered (eroded) within a 3 x 3 neighbourhood, and we have clarified this in the aforementioned sentence. The neighbourhood used here is the one surrounding each candidate cell, as delineated by moving away from the initial cell in Figure 2.4A in a direction opposite that cell's aspect. The red polygon in Figure 2.4 simply depicts the total extent of all 3 x 3 neighbourhoods created by this routine, and thus the cells where elevation will be adjusted (i.e., the cells which will undergo bank erosion). We have modified the title of the red polygon to read "Total Bank Erosion Extent" in the legend of Figure 2 in order to clarify this.*

In short, based on the description in the manuscript and associated figures, I would not be able to recreate the authors' bank erosion algorithm. I could, of course, look at their open source code to have all these things clarified, but that is beside the point.

Could the authors please expand the description of their bank erosion algorithm, or maybe provide a clearer illustration of its working?

*We have extensively modified the text in this section to better explain the bank erosion routine as described above.*

2e) During bank erosion when "eroded sediment is immediately transported downstream" (p11 ln 15), does that affect bed erosion capacity? In other words: is bed scour adjusted (reduced) when bank erosion occurs?

*Bank erosion/transport/deposition is computed separately from bed dynamics, and hydraulic modeling (i.e., Delft3D) is not re-run between these two steps; as a result, bed scour is not adjusted in areas nearby eroding banks. This obviously represents a simplification, as researchers have noted feedbacks between bank erosion and bed scour (see, for example, the review by Rinaldi and Darby, 2009), but this simplification was made here to conserve computational resources and to avoid re-computing the flow field twice over the course of a single event. This has been noted in the manuscript text.*

p6 ln 4: The mentions of (e.g., x) and (e.g., y) are redundant.

*These have been removed from the text.*

p6 ln 5: The phrase about rotation of the elevation model grid better fits with the description of the grid (p6 ln 12, 13)

*This text has been moved as suggested.*

p10 ln28: 60m2 --> 120 m2

*Changed to 120 $m^2$.*

p11 ln5: number of cells specified by Eq.(10) --> number of cells specified by Eq.(11).

*Corrected to Eq. 11*

The authors have indicated in their response to my earlier comment that there was nothing wrong with this equation reference. However, Eq.(10) does not specify a number of cells (it specifies a shear velocity), whilst Eq.(11) does.

*This reference should have been to Equation 11, and has been corrected. Thank you for pointing this out once again.*

p15 ln8: (Kasprak et al., 2017) is not included in the reference list

*We have now included this reference in the list*

p22 ln27: Delt3DThis --> Delft3D. This

*Corrected*

p28 ln7: et al --> et al.

*Inserted period in et al.*

all text: modeling --> modelling

*All instances of 'modeling' changed to 'modelling'*

(I believe EGU follows UK English spelling)

fig 2: It is somewhat confusing that subfigure 3A is placed next to subfigure 2. Intuitively one would expect each of the detail views to be associated with the workflow view to the left of it. It is for 1, 3B and 4, but not for 3A.

In their response to my earlier comment about this, the authors indicated that they agree, and have added arrows linking the sub-figures with their appropriate panels in Figure 3 to avoid confusion. This appears to not be the case.

*Our apologies for the oversight. We did add arrows to clarify the flow of the figure, but this updated figure was not initially included in the revised manuscript. We have ensured that this is now the case.*

figs 7-11: Text too small to read comfortably in many subpanels (axis labels, axis values, legends, scale bars)

*We appreciate the reviewer pointing this out, and where possible, we have increased font sizes in the relevant figures. That being said, the figures shown here are low-resolution JPEG format, embedded within the submitted MS Word document. We also have generated high-resolution vector versions of these files, and will ensure that they are published in the resultant manuscript pending its acceptance.*

fig 8: Include same subpanels as fig 7,9,11

*The majority of this information is found in Figure 7, including the hydrograph (Panel A), and field-based ECD and DoD (Panels B, D). As opposed to replicating this information in Figure 8, we have instead modified the figure caption to refer readers to the preceding Figure (7) for field survey results.*

**FROM REVIEWER #3**

The authors have done a generally good job of attending to previous review comments but there remain several issues with the present version that should be addressed. I will discuss those first, then list smaller/minor page by page things warranting corrections.

1. In the abstract (and later in the paper) the notion of representing hydrographs as a series of steady states is misleading – it continues to imply a duration associated with each steady state (note that I fully understand how time is removed from the calculations), and it implies that this expediency is at the root of the computational time saving over CFD type approaches. The implication is that by chopping event hydrographs down into smaller and smaller blocks the approach might converge on a CFD result. But that is never done in the paper – the example provided from the Rees is actually three events back to back – it is not a single event hydrograph split in 3 time blocks.

Indeed, a key (unstated) assumption of the approach is that each event modelled has adequate duration to allow the bedload to get to its destination – so you can see that breaking events up into smaller and smaller intervals will ultimately badly overcook the results, not give them greater precision. While I appreciate that this is never actually done, the paper hints that it could be.

*We did not intend to imply that a progressive dissection of hydrographs into sub-events (and modelling each of those) would converge on a CFD-like result. At its core, our rationale for pursuing an event-scale modeling approach was to examine whether such a simplified modelling strategy could produce behaviors seen in the field and produced by much more computationally-intensive and finer-timestep models. In Seciton 4.2.1., which deals with hydrograph discretization, we split a multi-day flood event into three discrete model runs in order to examine what, if any, changes in morphodynamics and braiding mechanisms would result from modeling notable sub-peaks within the event record, but we did not imply that continued downsampling of a hydrograph would eventually converge on a CFD-like solution. We have added text to this section and edited the text in the Introduction to avoid the appearance of implying this.*

*The second point above, that modeled events must be of adequate duration to allow bedload to reach its 'destination', as specified by the path-length distribution, is valid and we've addressed this in Section 2.4.*

2. On page 9 there are two things that niggle me. The first is that Montgomery's event averaged relation (eqn 7) is applied to the event peak discharge then assumed to be representative of the whole event – so all of sudden event averaged Qb becomes Qb at peak discharge, and ditto for bedload advection speed. What Eqn 7 is really predicting at the event peak is the active layer thickness. The second is that Eqn 8 is not the MPM formula – the scaling coefficient (8) is missing as are the density terms. The net result is that a factor of ~ 0.4 is missing from the equation. Hopefully this was an omission in the text and not the model.

*With regard to the former comment (Qb at peak discharge), it is certainly true that an alternative approach could be employed, which would involve computing the average discharge over the course of a flood and then computing the corresponding Qb. We did not explore this approach here, but we have suggested it as a potential future avenue in Section 2.2 For our purposes, the event peak discharge represented a simple metric for extracting hydraulic and sediment transport parameters across extended hydrograph records.*

*With regard to the latter comment (dimensionality of the MPM formula), this was an error in the text, and in fact we did not use the traditional MPM formula here, but rather employed a simple exponential relationship between excess bed shear stress and bedload transport rate; note that as compared to the MPM formula, this version is not dimensionless, instead using bed shear stress and not Shields stress. This was done for consistency with the original Montgomery et al. (1995) work, see their Equations 1 and 2, and because bed shear stress was easily obtained from Delft3D outputs. We argue that the absolute magnitude of bed scour depth is, in general, poorly known at the event scale, and that this approach simply represents a starting point at computing this quantity. We have noted the potential for improvements to Equation 7, and thus the event-scale scour component of the model, through future research.*

3. Section 2.5. The input boundary condition option used needs to specified for each model run. What was done in the cases of the first event in a series or for single events, like the first Rees example – since there is not a preceding event calculation?

Also, what was done at the lateral boundaries? It reads like bedload was allowed to diffuse across them but was not returned (e.g. on the opposite side) so the modelling should progressively lose sediment mass.

*The introduction/import of sediment to the reach occurs at the end of each event modelling run. This was necessary for the reasons the reviewer points out: in order to set imported sediment volume as a function of exported sediment volume, the amount of sediment leaving the reach must be known. This is illustrated*

*in Figure 1. Sediment leaving the lateral boundaries of the domain was also counted as being exported, and that volume, along with sediment leaving the downstream reach boundary, was included in the imported sediment volume.*

4. Two things on page 19. First, on line 18, the notion of altering the model timestep is misleading (as discussed in 1. above). The example used is actually a multipeak event, but is dominated by the big middle one. Note on line 20, 254 should be 259 (as elsewhere).

*We've corrected 254 m3/s to 259 m3/s, and have clarified that the model timestep remained at the event scale for each of the three sub-peaks.*

5. At the top of page 21, again, the assumption needs to be stated that the undisclosed event duration provides adequate time for the bedload to get to its destination. It would be a good idea to calculate this as an assumption check.

*We've again stated the transit-time assumption here.*

6. On page 22, there are potential implications of the upstream BC that require discussion. Around about line 12 it is important to say which upstream boundary condition is imposed. Also, I would like to see some discussion around how the sequencing of events impacts the results – for example, a large event before a small one will deliver a large wad of bedload into the top of the reach for the small event - promoting deposition there. It is mentioned that the top 25 m of the reach is not analysed but that is because of the hydraulic boundary condition issue. Surely the sequencing of events will affect the net apparent change between the first and last event in the Feshie sequence? I suggest a plot of mean bed level vs time to explore this?

*We have clarified that an equilibrium sediment budget was employed within this section. Further, we have added text to the Discussion regarding the potential role of flood sequencing on computed geomorphic change, while also noting that this is a topic deserving of further study throughout the literature. See Section 5.2.4.*

7. Section 5.1, which discusses emergent behaviour, really needs to discuss the emergent scour behaviour that develops. Given only a relatively small number of iterations (i.e. 185 events in the period 2003-2013), the tendency to scour, degrade, and simplify the channel network is quite notable (e.g. -8.07 elevation fall is very severe and quite unrealistic). While this is noted on page 22 (around line 25) with reference to the Singh et al Delft3D run, it doesn't mention that the Singh et al run also produced unrealistic results. So this issue appears to be being swept under the carpet whereas it really shows that the model in its present form is not to be trusted for predicting decadal or longer scale morphological evolution. While this is discussed some more on page 30, and explanations are offered, it seems pretty clear that the main channel has gotten into a (scour, increased flow capture, increased shear stress, increased ) feedback loop.

*We have added text to note this positive feedback loop at the end of Section 5.1., and have emphasized that further work on event-scale bed sediment scour will improve (and in this case, perhaps enable) model fidelity at extended timescales.*

P5, L 26: Need to say that Delft3D was run in 2D depth-averaged mode. Also, I suggest shifting the sentence on lines 9-12 of P6 to here.

*We note that Delft3D was run in two-dimensional depth averaged mode in the paragraph immediately following Equations 1 and 2; however, we have moved this text to the beginning of Section 2.1 as suggested.*

P7, L1-4: Needs some edits. Also, give units of m to the Ks values.

*We have edited some errors in this paragraph and clarified that Ks values are in m.*

P7, L11-12: Need to say at event timescales rather than coarser time intervals. The model is never being run at "coarser time intervals". The events are assumed to have undetermined duration. See also comment 1 above.

*Changed 'coarser' to 'event'*

P7, L26-27: Sorry, but I've never heard of a braided river being exhausted of bedload through a hydrograph. This might be reasonable for suspended load. I suspect latency issues created by the upstream bedload feed boundary condition is something better to evaluate (see 6 above).

*Fair point – we've removed this portion of the related sentence. This certainly applies to suspended sediment, but the model used here didn't investigate fine sediment dynamics.*

P8, L20: Needs to read Grad Qs. But also, why talk about Qs here then jump across to using Qb in Eqns 7 and 8?

*Updated to include $\nabla Qs$; we mention this here simply to illustrate the contrast between an Exner-based approach for computing bed elevation change and the approach taken in our event-scale model.*

P9, L 7: Need to say that Qb is the average unit mass-based bedload during the event.

*Changed to include 'mass-based'*

P 9, eqn 8: give the dimensionally correct version of the MP-M formula.

*See above – MPM formula was not used, but rather we employed a simple exponential relationship between dimensional bed shear stress and critical shear stress.*

P13, L17: Mode of what?

*Removed 'mode'*

P13, L20: Suggestion only, but I struggle to see that ST is particularly informative here, as it really weights the average sinuosity of individual channels (which is a useful measure of the morphological evolution) by the braiding index. In other words, if you use ST you don't know if it's due to a sinuosity change or a BI change.

*This is true, in that a change in the number of overall channels could change ST despite the sinuosity of individual channels remaining relatively constant. We believe that by including BI as well as ST, some of the potential ambiguity in ST changes can be overcome; as ST is included in numerous Figures (7, 9, 11), we've chosen to retain the metric here.*

P13, L28: It's not large differences in the metric, it's large non-zero values (and say what -values indicate and what +values indicate).

*Thanks for pointing this out – we've updated the text to reflect this.*

P15, L24: Say 0.5 m grid models – otherwise it suggests models accurate to 0.5 m in elevation.

*Changed to note that these are 0.5 m resolution DEMs.*

P16, L31: The model roughness parameter is the Nikuradse roughness length, ks.

*This is correct, in that the Nikuradse roughness length was used to estimate the Colebrook-White roughness, which was the parameter required for Delft3D model runs; see Equations 3 and 4.*

P17, all of section 4.1.1: This is confusing and needs to be written. On P 7 you say you calculate ks off grainsize, then here you first suggest you derived this by trial error (i.e. by calibration), then finally you say you just used Williams et al's calibration result. Is it that you used an initial grainsize-based value of ks, saw no reason to change this during calibration runs, and this aligned with what Williams et al got?

*We've rewritten this section for simplicity. While we did perform calibration of our Delft modeling on the Rees, we found that the values of roughness and eddy viscosity that produced hydraulic outputs most closely matching field data were identical to those found by Williams et al. (2013). It was simplest here to say that we just used the values obtained in that study, which is what's written now.*

P 20, L4: I'm curious – how much of the bed was mobile at 20 m3/s? Was that a good estimate based on the more detailed results coming from the model (which evaluates Tc everywhere)?

*Ashworth and Ferguson (1989) found that about 75% of the tracer particles with diameters of 0.1 m inserted into a study reach were mobilized at discharges around 20 $m^3$/s (see their Figure 4), but this does not speak to the amount of the bed that was mobile in an areal sense, and so we are unable to directly compare these estimates to our model outputs (which were run for a full year of competent discharges). This would certainly be an interesting additional layer of model validation for the future, however.*

P25, L6-8: It seems hardly surprising that the big 259 m3/s event dominated compared to the "pups" that came before and after it – particularly considering the threshold of motion influence. So I see no reason for the speculation that follows. I suggest deleting.

*We've deleted the relevant text.*

P27, L22: I disagree that it replicates the magnitude, rather, it produces similar.

*Changed to 'produce similar magnitudes'*

P27, L 24: It would be helpful to say if Williams et al thresholded their model results as done in this paper – so that that apples are being compared with apples.

*Yes, Williams et al. (2016b) also thresholded their model results at 0.1 m vertical change (see Table in that paper). This has been noted in the text.*

P32: I suggest acknowledging the referees!

*Included in Acknowledgements*

[revised manuscript text omitted]

15    **Figure 9.**

[Figure]

[Figure]

**Figure 10.**

[Figure]

**Figure 11.**

[Figure]

[Figure]

---

## Author Response (AR3)

**FROM REVIEWER #3**

Evaluation

The authors have addressed my main concerns from the previous iterations. I think the paper is publishable following a few minor corrections corresponding to a couple of minor issues, as listed below.

*We thank the reviewer for their time spent on this manuscript, which we believe has considerably improved in its quality and presentation. We have addressed the minor comments noted below.*

Minor issues

1) Be explicit about origin of model parameter values

The authors have not formally calibrated the model. But they have somehow qualitatively tuned some of their equations to "obtain the best possible agreement with areas of bank erosion delineated from successive DEMs on both the Feshie and the Rees Rivers" (author's response to one of my previous comments). I would like the authors to be a bit more explicit about this in the text. For example: What is the basis for setting k equal to 8 (p9 ln28)? Is it part of the same loose calibration, or are there other reasons? Similarly, explicitly mention that "Equation 11 is not physically based, and was derived through qualitative calibration, whereby the values used in Equation 11 are simply reflective of the best qualitative correspondence between modelled and field-derived estimates of areas undergoing bank retreat".

*We have inserted a paraphrased version of the suggested text on Page 12, at the end of Section 2.3., which clarifies that Equation 11 (bank erosion extent) is not a deterministic approach, but rather a simplified (and qualitatively-calibrated) means of estimating bank erosion extent to align with field observations, and one which uses readily-extracted values from field data and hydraulic modelling. The qualitative calibration was completed for bank erosion only, and not bed scour, which was determined through a widely-used excess shear stress approach. Equation 8 deals with bed sediment erosion (as opposed to bank erosion); the inclusion of k = 8 here was an error in the text; as noted in the previous review, Equation 8 is not the Meyer-Peter Mueller formula (which does contain this scaling factor), but rather a simple exponential (and dimensional) relationship between excess stress and bedload transport proposed by Montgomery et al. (1995). We have removed the '8' factor in Equation 8.*

2) Explicitly mention handling of sediment import for each experiment

The authors indicate that sediment import is user-specified and can be set in one of three ways (equal to the volume of sediment export during the preceding event; percent of sediment export during the preceding event; or specified via a text file sedigraph timeseries) (p14 ln18). However, it is not clear which of those three is used for the simulations described in the paper. Section 4.3 (particle length case study) contains an explicit mention that "in all simulations we employed an equilibrium sediment budget" (p23 ln13), but it is not clear if this applies to all simulations in the particle length case study, or if this applies to all other case studies too. Presumably the other case studies (Sections 4.1., 4.2, 4.4) did not apply a equilibrium sediment budget since they were able to compute net volumetric erosion (under equilibrium this would not happen: volumetric erosion would equal volumetric deposition, and the net difference would be zero). In any case, please explicitly state how sediment import was handled in each of the case studies.

*We have clarified that an equilibrium sediment budget was used in Sections 4.2, 4.2.1, and 4.4. Note that small divergence in the volume of erosion and deposition (i.e., a net depositional or erosional volumetric sediment budget) is possible even under equilibrium conditions as a result of (a) vertical thresholding, which preferentially removes low-magnitude changes, often resulting from depositional processes, and (b) rounding of very small amounts of deposition and erosion during sediment import; see Figure 3. The more pronounced divergence in the volumetric erosion and deposition components of the budget in the decadal-scale modelling (Section 4.4.) are due to the analysis mask used in computation of geomorphic change, which excluded the upstream-most 25 m of the reach.*

p13 ln9:          e.g. --> i.e.

*Changed 'e.g.' to 'i.e.'*

p13 ln12:          e.g. --> i.e.

*Changed 'e.g.' to 'i.e.'*

p30 ln 30:          three discrete events --> multiple discrete events

*Changed 'discrete' to 'multiple'*

Fig 7: The first four entries in the data table under subpanel A contain numbers in brackets. Please add an explanation to the figure caption to indicate what these mean

*These values reflect the change in braiding index and total sinuosity that occurred over the course of the event (or year, or decade in the case of Figures 9 or 11, respectively). We have updated the Figure 7 caption to reflect this.*

Fig 8: In an earlier review I had requested that this figure be laid out in the same style as Figs 7, 9 and 11. The authors are reluctant to do so because of duplication of information already contained in subpanels A,B and D from Figure 7. Instead, they provide rather focus on new information equivalent to what is shown in subpanels C and E from figure 7. However, in doing so, they omit the equivalent information equivalent to subpanels F and G, i.e. they omit the spatial map of modelled braiding mechanisms (F), and their volumetric contributions in relation to the observed mechanisms (G). I think this information would be useful, to see how much or how little of a difference the 3-point discretization makes to the attribution of braiding mechanisms, relative to the 1-point discretization. Can the authors please add an equivalent to subpanels F and G to Figure 8?

I still think that the easiest and most consistent way to do this is to follow the same layout as Fig 7, 9, 11, even if that duplicates some of the information displayed – but I'm not too bothered if they authors prefer a different layout.

*Our apologies for this oversight – we have now included a bar plot panel, identical to those in Figures 7, 9, and 11, showing braiding mechanisms from the discretized case study on the Rees (Panel D, Figure 8). We feel that simply presenting the bar plot rather than this and the mapped mechanisms provides a simpler, more balanced figure layout. Additionally, the braiding mechanisms' contribution and spatial location was very similar to the single-event run on the Rees.*

[revised manuscript text omitted]

*MODELED DoD*

[Figure]

*MODEL ECD*

[Figure]

**Figure 9.**

[Figure]

**Figure 10.**

[Figure]

**Figure 11.**

[Figure]

185 modeled peaks > 20 m³/s

IB (Field): **2.4** **(+0.6)**
IB (Model): **3.8** **(+2.0)**
ST (Field): **2.9** **(+0.4)**
ST (Model): **3.6** **(+1.1)**
Con: **96** Diff: **27** CH: **63** (Field)
Con: **55** Diff: **20** CH: **36** (Model)

Central Bar Development
Chute Cutoff
Lobe Dissection
Transverse Bar Conversion
Bank Erosion

Bar Edge Trimming
Channel Incision
Confluence Pool Scour
Lateral Bar Development
Overbank Sheets

*FIELD*

*MODEL*

*FIELD DoD*   *MODELED DoD*   *BRAIDING MECHANISMS*

Feshie Braiding Mechanisms, 2003 - 2013

---

## Author Response (AR4)

*Changes from Associate Editor Turowski:*

14.30

Fixed spelling of 'Mohrig"

22.29

Removed 'strikingly'

27.11

Reworded to remove 'striking'

[revised manuscript text omitted]